# Multi-Objective Optimization of Functionally Graded Beams Using a Genetic Algorithm with Non-Dominated Sorting

Chih-Ping Wu *[ID] and Kuan-Wei Li

Department of Civil Engineering, National Cheng Kung University, Tainan 70101, Taiwan; wylj966190@gmail.com
* Correspondence: cpwu@mail.ncku.edu.tw

**Abstract:** A mixed layer-wise (LW) higher-order shear deformation theory (HSDT) is developed for the thermal buckling analysis of simply-supported, functionally graded (FG) beams subjected to a uniform temperature change. The material properties of the FG beam are assumed to be dependent on the thickness and temperature variables, and the effective material properties are estimated using either the rule of mixtures or the Mori–Tanaka scheme. The results shown in the numerical examples indicate the mixed LW HSDT solutions for critical temperature change parameters are in excellent agreement with the accurate solutions available in the literature. A multi-objective optimization of FG beams is presented to maximize the critical temperature change parameters and to minimize their total mass using a non-dominated sorting-based genetic algorithm. Some specific forms for the volume fractions of the constituents of the FG beam are assumed in advance, such as the one- and three-parameter power-law functions. The former is used in the thermal buckling analysis of the FG beams for comparison purposes, and the latter is used in their optimal design.

**Keywords:** functionally graded beams; genetic algorithms; layer-wise beam theories; multi-objective optimization; non-dominated sorting; thermal buckling



## 1. Introduction

Functionally graded (FG) beams, plates, and shells are emerging heterogeneous material structures, which are formed by mixing two- or multiple-phase materials with a pre-designed spatial distribution of volume fractions of the constituents [1–4]. Because the material properties of FG structures gradually and smoothly vary through their domain, some drawbacks can be prevented, including delamination and stress concentration, which usually occur at the interfaces between adjacent layers in the case of the laminated composite structures due to the material properties suddenly changing at these locations. On the other hand, FG structures provide a large optimal design space in engineering practice. Engineers can design the spatial distributions of the volume fractions of the constituents according to practice demands to obtain the best structural physical properties. Based on the above-mentioned advantages of FG structures, they are also becoming increasingly more popular in various high-end industries, such as aerospace, submarine, automobile, and nuclear industries, as well as for use in biomedical implants [5,6]. A variety of mechanical analyses of FG structures and single/multi-objective optimization of these structures, thus, are attracting considerable attention.

Some comprehensive literature surveys with regard to articles examining the mechanical analyses of FG structures can be found in the public literature [7–14]. Among these, the literature survey in this work focuses on articles investigating the thermal buckling behavior of FG beams using various beam theories and on articles related to the single- and multi-objective optimizations of FG beams.

Based on the Euler–Bernoulli theory (EBT), Kiani and Eslami [15] examined the thermal buckling behavior of FG beams under different boundary conditions, in which only the bending effects were considered in their analysis, rather than the shear deformation

effects. The first-order shear deformation theory (FSDT) was used by Tauchert [16] to study the thermal buckling of simply supported, antisymmetric angle-ply laminates, in which the shear deformations of the laminates were simply assumed as a constant through their thickness direction. Reddy [17] developed a refined shear deformation theory (RSDT) to investigate the various mechanical behaviors of laminated composite (LC) structures. In Reddy's formulation, the shear deformations induced in the LC structures were assumed to be a parabolic function distribution through the thickness direction; the shear correction factor commonly used in the FSDT was not needed, and the surface conditions regarding the traction stresses free were satisfied. In order to more precisely capture the shear deformation effects on the various mechanical behaviors of LC and FG structures, some advanced shear deformation theories have been proposed, including the third-order shear deformation theory (TSDT) [18,19], the sinusoidal shear deformation theory (SSDT) [20,21], the hyperbolic refined shear deformation theory (HRSDT) [22,23], and the exponential shear deformation theory (ESDT) [24]. Liu et al. [25] investigated the nonlinear response of FG structures using an iso-geometric continuum shell element method. Liu et al. [26] presented an analysis of thermo-mechanical snap-through instability of structures using a hybrid load- and displacement-controlled algorithm. Based on a shear deformation theory, Refrafi et al. [27] examined the hygro-thermo-mechanical buckling behavior of simply-supported FG sandwich plates resting on a Winkler–Pasternak foundation. Based on Carrera's unified formulation (CUF) [28], the above-mentioned advanced and refined shear deformation theories can be hierarchically derived and included as special cases [29]. Some comparative studies for different mechanical behavior analyses of FG beams using assorted advanced and refined shear deformation beam theories have been carried out, including thermal buckling and post-buckling analyses [30], bending and free vibration analyses [31], thermal buckling analysis [32], and free vibration analyses [33,34]. The above-mentioned theories can be summarized as equivalent-single-layered (ESL) theories, in which a global coordinate system was introduced to describe the displacement field induced in LC and FG structures.

Because ESL theories are often inadequate for determining the three-dimensional (3D) stress field induced in loaded LC and FG beams, some layer-wise (LW) theories for these beams have been presented. Based on Hamilton's theory, Tahani [35] developed a displacement-based LW theory for the static bending and free vibration analyses of LC beams. Lee and Saravanos [36] developed an LW FSDT for coupled thermo-piezo-electric composite beams under thermal loads. Shimpi and Ainapure [37] and Shimpi and Ghugal [38] developed an LW trigonometric shear deformation theory for the static analysis of LC beams. Based on an LW higher-order shear deformation theory (HSDT), Pandey and Pradyumna [39] developed a finite element method (FEM) for the static and dynamic analyses of FG sandwich plates. Bayat and Toussi [40] carried out thermal buckling and thermal postbuckling analyses of LC beams reinforced with shape memory alloy wires using a displacement-based LW theory. Based on the stationary principle of minimum potential energy [41,42] combined with the Lagrange multiplier method, Wu and Kuo [43] and Wu and Chen [44] developed a mixed LW HSDT for the bending, free vibration, and buckling behavior of LC plates. Wu and Xu [45] extended this approach to develop the strong and weak formulations of a mixed higher-order shear deformation theory for the static analysis of FG beams under thermo-mechanical loads. A comprehensive overview with regard to the development, numerical implementation, and application of LW theories was undertaken by Liew et al. [46].

Some articles related to the optimal design of LC structures and FG beams have been presented using different single-and multi-objective optimization algorithms combined with advanced and refined shear deformation beam theories. Walker et al. [47,48] developed an optimal procedure to select the best material combinations for minimum weight and cost of hybrid composite plates and for minimum mass of sandwiched composite cylindrical shells, respectively. Houmat [49] and Guenanou and Houmat [50] proposed a layer-wise optimization method to obtain an optimal lay-up design for maximum fun-

damental frequency of variable stiffness LC plates and symmetrically LC circular plates, respectively. Cho and Ha [51] presented volume fraction optimization for the purpose of minimizing thermal stresses induced in Ni-Al$_2$O$_3$ FG beams. Based on the Vlasov thin-walled theory [52], Nguyen and Lee [53] committed to the optimal design of thin-walled FG beams for buckling problems, in which an LW cubic interpolation for the through-thickness distribution of the volume fraction of the constituents was assumed, and a genetic algorithm (GA) was used as the optimal tool [54,55]. Kamarian et al. [56] studied the optimization of material compositions of FG beams to maximize their fundamental natural frequencies using the firefly algorithm and an adaptive neuro-fuzzy inference system. Yas et al. [57,58] engaged in the optimization of material compositions of a three-parameter FG beam in order to maximize its fundamental frequencies using the imperialist competitive algorithm, an artificial neural network [59], and a generalized differential quadrature (DQ) method [60–62]. Goupee and Vel [63] proposed a methodology to maximize the fundamental frequencies of FG structures by tailoring their material distributions. Na and Kim [64–66] investigated the volume fraction optimization of FG plates and panels under thermal loads to minimize the induced thermal stresses and to maximize the critical temperature parameters, in which a bi-objective optimization was considered and the optimal tool was a quasi-Newton method. Walker and Smith [67] presented a multi-objective optimization technique for minimizing the weighted sum of the mass and deflection of LC structures, in which a GA and an FEM were used. Tornabene and Ceruti [68] dealt with mixed static and dynamic optimization of four-parameter FG structures. The FSDT combined with the generalized DQ method was used for the static and dynamic analyses of the FG structures, and three different optimization schemes were used as the optimal tools, including the particle swarm optimization, the Monte Carlo, and the GA approaches.

In this work, the authors aim at investigating the material composition optimization of an FG beam under a uniform temperature change in order to maximize the critical temperature change parameter and minimize the total mass of the FG beam. A mixed LW HSDT [45] is developed for the thermal buckling analysis of the FG beam. The current solutions for the critical temperature change parameters of the FG beam obtained through considering the temperature-dependent (TD) and the temperature-independent (TI) material properties are examined, and solutions obtained using the von Kármán geometric nonlinear strains (GNS) and the full GNS are also estimated. A non-dominated sorting-based GA [69,70] is used for the current multi-objective optimization analysis in the current issue. The through-thickness distribution of the material properties of the FG beam is assumed to be a three-parameter power-law function of the volume fractions of the constituents [71,72], the material-property gradient indices of which are, thus, to be determined for the optimal material profile.

## 2. Effective Material Properties

In this work, two micromechanics models, the rule of mixtures [73,74] and the Mori–Tanaka scheme [75], are used to estimate the effective material properties of the FG beam, as described in the following sections.

### 2.1. The Rule of Mixtures

According to the rule of mixtures, the through-thickness distributions of the effective material properties of the FG beam are written in the following form,

$$F_{eff}(z) = \Gamma_c(z)\, F_c + \Gamma_m(z)\, F_m = F_m + (F_c - F_m)\, \Gamma_c(z), \tag{1}$$

where $\Gamma_c$ and $\Gamma_m$ represent the volume fractions of the ceramic and metal materials of the constituents of the FG beam, respectively, such that $\Gamma_c + \Gamma_m = 1$. $F$ can be one of the engineering constants, including Young's modulus $E$, Poisson's ratio $v$, thermal expansion coefficient $\alpha$, and mass density $\rho$. The subscripts $m$ and $c$ represent the metal material and the ceramic material, respectively.

### 2.2. The Mori-Tanaka Scheme

Within the frame work of Eshelby's elastic inclusion theory [73,74] restricted to a single inclusion in a semi-infinite elastic, homogeneous and isotropic medium, Mori and Tanaka [75] extended the original theory to more general cases of multiple inclusions embedded into a finite elastic medium, in which an equivalent inclusion-average stress method was used.

For a two-phase isotropic material, the effective material properties are written as the explicit formula [76,77] as follows:

$$K(z) = \Gamma_c \left(K_c - K_m\right) / \left[1 + (1 - \Gamma_c)\left(K_c - K_m\right) / \left(K_m + (4/3)\,G_m\right)\right] + K_m, \tag{2}$$

$$G(z) = \Gamma_c \left(G_c - G_m\right) / \left[1 + (1 - \Gamma_c)\left(G_c - G_m\right) / \left(G_m + f_m\right)\right] + G_m, \tag{3}$$

$$\alpha(z) = \Gamma_c \left(\alpha_c - \alpha_m\right) / \left\{K(z)\left[1 - \left(K(z) - K_c\right) / \left(K(z) - K_m\right)\right]\right\} + \alpha_m, \tag{4}$$

in which $f_m = G_m(9K_m + 8G_m) \, / \left[6\left(K_m + 2G_m\right)\right]$, and $K$ and $G$ represent the volume and the shear moduli, respectively.

In later sections of this paper, if no specific mention is given the effective material properties of the FG beam are estimated using the rule of mixtures.

## 3. The Mixed LW HSDT for FG Beams

In this work, the authors develop a unified formulation of the mixed LW HSDT for the thermal buckling analysis of simply supported FG beams subjected to a uniform temperature change, in which the material properties of the FG beam are considered to be temperature-dependent. The configuration and coordinates of the FG beam are shown in Figure 1a, in which $h$ and $L$ represent the thickness and the length of the FG beam, respectively. In the analysis, the FG beam is artificially divided into $N_l$ layers as shown in Figure 1b, and the thickness of each individual layer constituting the beam is $h_m$ $(m = 1 - N_l)$, such that $\sum\limits_{m=1}^{N_l} h_m = h$.

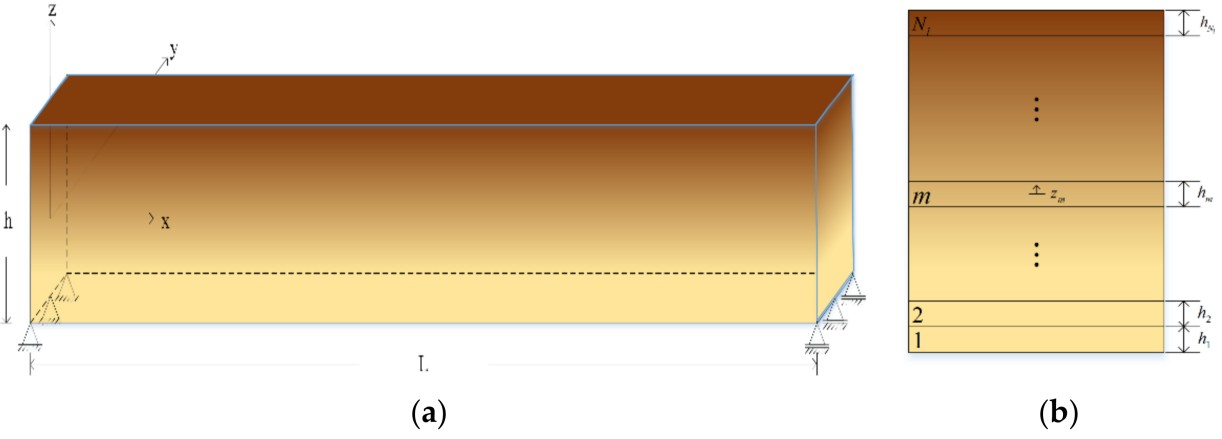

**Figure 1.** (**a**) Configuration and coordinates of a simply-supported functionally graded (FG) beam; (**b**) cross section of an $N_l$-layered FG beam.

### 3.1. Strong-Form Formulation

In the current LW HSDT, the displacement field for a typical individual layer is given as follows:

$$u^{(m)}(x, z_m) = u_0^{(m)}(x) + z_m\, u_1^{(m)}(x) + z_m^2\, u_2^{(m)}(x) + z_m^3\, u_3^{(m)}(x), \tag{5}$$

$$w^{(m)}(x, z_m) = w_0^{(m)}(x) + z_m\, w_1^{(m)}(x) + z_m^2\, w_2^{(m)}(x) + z_m^3\, w_3^{(m)}(x), \tag{6}$$

where $m = 1, 2, \ldots, N_l$; $u_0^{(m)}$ and $w_0^{(m)}$ denote the mid-plane displacements of the layer in the $x$ and $z$ directions, and their $i$th-order expansion terms along the local thickness direction are $u_i^{(m)}$ and $w_i^{(m)}$ ($i = 1, 2,$ and $3$). The displacement components in the $y$ direction are taken to be zero.

According to the perfect bonding assumptions at the interfaces between adjacent layers, the corresponding displacement continuity conditions at these places are given as

$$
\begin{aligned}
f_x^{(k)} &= \left[ u_0^{(k+1)} - (h_{k+1}/2)\, u_1^{(k+1)} + \left( h_{k+1}^2/4 \right) u_2^{(k+1)} - \left( h_{k+1}^3/8 \right) u_3^{(k+1)} \right] \\
&\quad - \left[ u_0^{(k)} + (h_k/2)\, u_1^{(k)} + \left( h_k^2/4 \right) u_2^{(k)} + \left( h_k^3/8 \right) u_3^{(k)} \right] \\
&= 0,
\end{aligned}
\tag{7}
$$

$$
\begin{aligned}
f_z^{(k)} &= \left[ w_0^{(k+1)} - (h_{k+1}/2)\, w_1^{(k+1)} + \left( h_{k+1}^2/4 \right) w_2^{(k+1)} - \left( h_{k+1}^3/8 \right) w_3^{(k+1)} \right] \\
&\quad - \left[ w_0^{(k)} + (h_k/2)\, w_1^{(k)} + \left( h_k^2/4 \right) w_2^{(k)} + \left( h_k^3/8 \right) w_3^{(k)} \right] \\
&= 0,
\end{aligned}
\tag{8}
$$

where $k = 1, 2, \ldots, (N_l - 1)$.

The strain-displacement relationship is given as

$$
\begin{aligned}
\varepsilon_x^{(m)} &= \left( \varepsilon_x^{(m)} \right)_l + \left( \varepsilon_x^{(m)} \right)_{nl} \\
&= \left( u_0^{(m)},_x + z_m\, u_1^{(m)},_x + z_m^2\, u_2^{(m)},_x + z_m^3\, u_3^{(m)},_x \right) \\
&\quad + (1/2) \left[ \left( u_0^{(m)},_x + z_m\, u_1^{(m)},_x + z_m^2\, u_2^{(m)},_x + z_m^3\, u_3^{(m)},_x \right)^2 + \left( w_0^{(m)},_x + z_m\, w_1^{(m)},_x + z_m^2\, w_2^{(m)},_x + z_m^3\, w_3^{(m)},_x \right)^2 \right],
\end{aligned}
\tag{9}
$$

$$
\begin{aligned}
\varepsilon_z^{(m)} &= w^{(m)},_{z_m} \\
&= w_1^{(m)} + 2z_m\, w_2^{(m)} + 3z_m^2\, w_3^{(m)},
\end{aligned}
\tag{10}
$$

$$
\begin{aligned}
\gamma_{xz}^{(m)} &= u^{(m)},_{z_m} + w^{(m)},_x \\
&= \left( w_0^{(m)},_x + u_1^{(m)} \right) + z_m \left( w_1^{(m)},_x + 2u_2^{(m)} \right) + z_m^2 \left( w_2^{(m)},_x + 3u_3^{(m)} \right) + z_m^3 \left( w_3^{(m)},_x \right),
\end{aligned}
\tag{11}
$$

where the commas denote the derivative of the suffix variable, and remaining strains are zeroes, including $\varepsilon_y^{(m)}$, $\gamma_{yz}^{(m)}$, and $\gamma_{xy}^{(m)}$. $\left( \varepsilon_x^{(m)} \right)_{nl}$ denotes the full GNS, and when the von Kármán GNS is considered, it is reduced as $\left( \varepsilon_x^{(m)} \right)_{nl} = (1/2) \left( w_0^{(m)},_x + z_m\, w_1^{(m)},_x + z_m^2\, w_2^{(m)},_x + z_m^3\, w_3^{(m)},_x \right)^2$. It is noted that the displacement continuity conditions at the interfaces between adjacent layers are imposed in this formulation, while the strain continuity conditions at these places are discontinuous.

The stress-strain relationship for an orthotropic material in thermal environment is given as

$$
\left\{ \begin{array}{c} \sigma_x^{(m)} \\ \sigma_z^{(m)} \\ \tau_{xz}^{(m)} \end{array} \right\} = \left[ \begin{array}{ccc} Q_{11}^{(m)} & Q_{13}^{(m)} & 0 \\ Q_{13}^{(m)} & Q_{33}^{(m)} & 0 \\ 0 & 0 & Q_{55}^{(m)} \end{array} \right] \left\{ \begin{array}{c} \varepsilon_x^{(m)} \\ \varepsilon_z^{(m)} \\ \gamma_{xz}^{(m)} \end{array} \right\} - \left\{ \begin{array}{c} Q_{1\alpha}^{(m)} \\ Q_{3\alpha}^{(m)} \\ 0 \end{array} \right\} \Delta T,
\tag{12}
$$

where the symbol $\Delta T$ denotes the temperature change, measured from a room temperature of 300K. $Q_{11}^{(m)} = \left[ (1 - v_{23}v_{32})/(E_2 E_3 \Delta) \right]^{(m)}$, $Q_{13}^{(m)} = \left[ (v_{13} + v_{12}v_{32})/(E_1 E_2 \Delta) \right]^{(m)}$, $Q_{33}^{(m)} = \left[ (1 - v_{12}v_{21})/(E_1 E_2 \Delta) \right]^{(m)}$, $\qquad Q_{55}^{(m)} = G_{13}^{(m)}$, $\Delta = \left[ (1 - v_{12}v_{21} - v_{23}v_{32} - v_{31}v_{13} - 2v_{21}v_{32}v_{13})/(E_1 E_2 E_3) \right]^{(m)}$, $Q_{1\alpha}^{(m)} = Q_{11}^{(m)} \alpha_1^{(m)} + Q_{13}^{(m)} \alpha_3^{(m)}$, $Q_{3\alpha}^{(m)} = Q_{13}^{(m)} \alpha_1^{(m)} + Q_{33}^{(m)} \alpha_3^{(m)}$, in which the subscripts 1, 2, and 3 denote the principle axes of the material properties, and that $E$, $v$, *and* $\alpha$ represent the

Young's modulus, Poisson's ratio, and the thermal expansion coefficients. For isotropic materials, these stiffness coefficients will be reduced to $Q_{11}^{(m)} = Q_{33}^{(m)} = \{E(1-v)/[(1+v)(1-2v)]\}^{(m)}$, $Q_{13}^{(m)} = \{Ev/[(1+v)(1-2v)]\}^{(m)}$, $Q_{55}^{(m)} = \{E/[2(1+v)]\}^{(m)}$, and $\alpha_1^{(m)} = \alpha_3^{(m)} = \alpha$. Note that these engineering constants $E$, $v$, and $\alpha$ in the analysis are considered to be dependent from the thickness and the temperature variables, and $\Delta T$ is a constant. In addition, other stress components are zero, including $\sigma_y^{(m)}$, $\tau_{yz}^{(m)}$, and $\tau_{xy}^{(m)}$, because a two-dimensional beam model is used.

The governing equations and associated boundary conditions are derived using the stationary principle of minimum potential energy combined with the Lagrange multiplier method, in which the displacement continuity conditions at the interfaces between adjacent layers given in Equations (7) and (8) are multiplied by the Lagrange multipliers and then substituted into the potential energy functional as the constraints, such that the extended potential energy functional of the $N_l$-layered FG beam is given, as follows:

$$
\begin{aligned}
\Pi_p = &\sum_{m=1}^{N_l} \int_0^L \int_{-h_m/2}^{h_m/2} \left[ (1/2)\,\sigma_x^{(m)}\,\varepsilon_x^{(m)} + (1/2)\,\sigma_z^{(m)}\,\varepsilon_z^{(m)} + (1/2)\,\tau_{xz}^{(m)}\,\gamma_{xz}^{(m)} \right] dz_m\, dx \\
&- (\Delta T) \sum_{m=1}^{N_l} \int_0^L \int_{-h_m/2}^{h_m/2} Q_{1\alpha}^{(m)} \left( \varepsilon_x^{(m)} \right)_{nl} dz_m dx + \sum_{k=1}^{(N_l-1)} \int_0^L \left[ \left( \lambda_x^{(k)} \right)\left( f_x^{(k)} \right) + \left( \lambda_z^{(k)} \right)\left( f_z^{(k)} \right) \right] dx,
\end{aligned}
\tag{13}
$$

where $\lambda_x^{(m)}$ and $\lambda_z^{(m)}$ are the Lagrange multipliers, which are identical to the transverse shear and normal stress components (i.e., $\tau_{xz}^{(m)}$ and $\sigma_z^{(m)}$) induced at the interfaces between adjacent layers, respectively.

Applying the stationary principle of minimum potential energy, following a standard variational process, and integrating the stress variables through the thickness direction, the authors obtain

$$
\begin{aligned}
\delta\Pi_p = &\sum_{m=1}^{N_l} \int_0^L \Big\{ \left( N_x^{(m)} - N_{xt}^{(m)} \right)\delta u_0^{(m)}{}_{,x} + \left( M_x^{(m)} - M_{xt}^{(m)} \right)\delta u_1^{(m)}{}_{,x} + \left( P_x^{(m)} - P_{xt}^{(m)} \right)\delta u_2^{(m)}{}_{,x} + \left( R_x^{(m)} - R_{xt}^{(m)} \right)\delta u_3^{(m)}{}_{,x} \\
&+ \left( N_z^{(m)} - N_{zt}^{(m)} \right)\delta w_1^{(m)} + 2\left( M_z^{(m)} - M_{zt}^{(m)} \right)\delta w_2^{(m)} + 3\left( P_z^{(m)} - P_{zt}^{(m)} \right)\delta w_3^{(m)} \\
&+ N_{xz}^{(m)} \left( \delta w_0^{(m)}{}_{,x} + \delta u_1^{(m)} \right) + M_{xz}^{(m)} \left( \delta w_1^{(m)}{}_{,x} + 2\delta u_2^{(m)} \right) + P_{xz}^{(m)} \left( \delta w_2^{(m)}{}_{,x} + 3\delta u_3^{(m)} \right) + R_{xz}^{(m)} \left( \delta w_3^{(m)}{}_{,x} \right) \Big\} dz_m\, dx \\
&+ (\Delta T) \sum_{m=1}^{N_l} \int_0^L \Big\{ S_1^{(m)}\,\delta u_0^{(m)} + S_2^{(m)}\delta u_1^{(m)} + S_3^{(m)}\delta u_2^{(m)} + S_4^{(m)}\delta u_3^{(m)} + S_5^{(m)}\delta w_0^{(m)} + S_6^{(m)}\delta w_1^{(m)} + S_7^{(m)}\delta w_2^{(m)} + S_8^{(m)}\delta w_3^{(m)} \Big\} dx \\
&+ \sum_{k=1}^{(N_l-1)} \int_0^L \left[ \left( \delta\lambda_x^{(k)} \right)\left( f_x^{(k)} \right) + \left( \delta\lambda_z^{(k)} \right)\left( f_z^{(k)} \right) + \left( \lambda_x^{(k)} \right)\left( \delta f_x^{(k)} \right) + \left( \lambda_z^{(k)} \right)\left( \delta f_z^{(k)} \right) \right] dx + (\text{boundary terms}) = 0,
\end{aligned}
\tag{14}
$$

where
$$
\begin{bmatrix} N_x^{(m)} & N_z^{(m)} & N_{xz}^{(m)} \\ M_x^{(m)} & M_z^{(m)} & M_{xz}^{(m)} \\ P_x^{(m)} & P_z^{(m)} & P_{xz}^{(m)} \end{bmatrix} = \int_{-h_m/2}^{h_m/2} \begin{Bmatrix} 1 \\ z_m \\ z_m^2 \end{Bmatrix} \begin{Bmatrix} \sigma_x^{(m)} & \sigma_z^{(m)} & \tau_{xz}^{(m)} \end{Bmatrix} dz_m, \begin{Bmatrix} R_x^{(m)} & R_{xz}^{(m)} \end{Bmatrix} = \int_{-h_m/2}^{h_m/2} z_m^3 \begin{Bmatrix} \sigma_x^{(m)} & \tau_{xz}^{(m)} \end{Bmatrix} dz_m,
$$

$$
\begin{bmatrix} N_{xt}^{(m)} & N_{zt}^{(m)} \\ M_{xt}^{(m)} & M_{zt}^{(m)} \\ P_{xt}^{(m)} & P_{zt}^{(m)} \end{bmatrix} = \int_{-h_m/2}^{h_m/2} \begin{Bmatrix} 1 \\ z_m \\ z_m^2 \end{Bmatrix} \begin{Bmatrix} Q_{1\alpha}^{(m)}\Delta T & Q_{3\alpha}^{(m)}\Delta T \end{Bmatrix} dz_m, R_{xt}^{(m)} = \int_{-h_m/2}^{h_m/2} z_m^3 \left( Q_{1\alpha}^{(m)}\Delta T \right) dz_m,
$$

$$
S_1^{(m)} = A_{1\alpha}^{(m)} u_0^{(m)}{}_{,xx} + B_{1\alpha}^{(m)} u_1^{(m)}{}_{,xx} + D_{1\alpha}^{(m)} u_2^{(m)}{}_{,xx} + F_{1\alpha}^{(m)} u_3^{(m)}{}_{,xx},
$$

$$
S_2^{(m)} = B_{1\alpha}^{(m)} u_0^{(m)}{}_{,xx} + D_{1\alpha}^{(m)} u_1^{(m)}{}_{,xx} + F_{1\alpha}^{(m)} u_2^{(m)}{}_{,xx} + H_{1\alpha}^{(m)} u_3^{(m)}{}_{,xx},
$$

$$
S_3^{(m)} = D_{1\alpha}^{(m)} u_0^{(m)}{}_{,xx} + F_{1\alpha}^{(m)} u_1^{(m)}{}_{,xx} + H_{1\alpha}^{(m)} u_2^{(m)}{}_{,xx} + J_{1\alpha}^{(m)} u_3^{(m)}{}_{,xx},
$$

$$
S_4^{(m)} = F_{1\alpha}^{(m)} u_0^{(m)}{}_{,xx} + H_{1\alpha}^{(m)} u_1^{(m)}{}_{,xx} + J_{1\alpha}^{(m)} u_2^{(m)}{}_{,xx} + L_{1\alpha}^{(m)} u_3^{(m)}{}_{,xx},
$$

$$
S_5^{(m)} = A_{1\alpha}^{(m)} w_0^{(m)}{}_{,xx} + B_{1\alpha}^{(m)} w_1^{(m)}{}_{,xx} + D_{1\alpha}^{(m)} w_2^{(m)}{}_{,xx} + F_{1\alpha}^{(m)} w_3^{(m)}{}_{,xx},
$$

$$
S_6^{(m)} = B_{1\alpha}^{(m)} w_0^{(m)}{}_{,xx} + D_{1\alpha}^{(m)} w_1^{(m)}{}_{,xx} + F_{1\alpha}^{(m)} w_2^{(m)}{}_{,xx} + H_{1\alpha}^{(m)} w_3^{(m)}{}_{,xx},
$$

$$
S_7^{(m)} = D_{1\alpha}^{(m)} w_0^{(m)}{}_{,xx} + F_{1\alpha}^{(m)} w_1^{(m)}{}_{,xx} + H_{1\alpha}^{(m)} w_2^{(m)}{}_{,xx} + J_{1\alpha}^{(m)} w_3^{(m)}{}_{,xx},
$$

$$S_8^{(m)} = F_{1\alpha}^{(m)} w_0^{(m)},_{xx} + H_{1\alpha}^{(m)} w_1^{(m)},_{xx} + J_{1\alpha}^{(m)} w_2^{(m)},_{xx} + L_{1\alpha}^{(m)} w_3^{(m)},_{xx},$$

$$\left\{ A_{1\alpha}^{(m)} \quad B_{1\alpha}^{(m)} \quad D_{1\alpha}^{(m)} \quad F_{1\alpha}^{(m)} \quad H_{1\alpha}^{(m)} \quad J_{1\alpha}^{(m)} \quad L_{1\alpha}^{(m)} \right\} = \int_{-h_m/2}^{h_m/2} Q_{1\alpha}^{(m)} \left\{ 1 \quad z_m \quad z_m^2 \quad z_m^3 \quad z_m^4 \quad z_m^5 \quad z_m^6 \right\} dz_m.$$

Performing Equation (14) the integration by part yields

$$
\begin{aligned}
\delta \Pi_p = &\sum_{m=1}^{N_l} \int_0^L \Big\{ -\left(N_x^{(m)} - N_{xt}^{(m)}\right),_x \delta u_0^{(m)} - \left(M_x^{(m)} - M_{xt}^{(m)}\right),_x \delta u_1^{(m)} - \left(P_x^{(m)} - P_{xt}^{(m)}\right),_x \delta u_2^{(m)} - \left(R_x^{(m)} - R_{xt}^{(m)}\right),_x \delta u_3^{(m)} \\
&+ \left(N_z^{(m)}, -, N_{zt}^{(m)}\right) \delta w_1^{(m)} + 2\left(M_z^{(m)} - M_{zt}^{(m)}\right) \delta w_2^{(m)} + 3\left(P_z^{(m)} - P_{zt}^{(m)}\right) \delta w_3^{(m)} \\
&- N_{xz}^{(m)},_x \delta w_0^{(m)} + N_{xz}^{(m)} \delta u_1^{(m)} - M_{xz}^{(m)},_x \delta w_1^{(m)} + 2M_{xz}^{(m)} \delta u_2^{(m)} - P_{xz}^{(m)},_x \delta w_2^{(m)} + 3P_{xz}^{(m)} \delta u_3^{(m)} - R_{xz}^{(m)},_x \delta w_3^{(m)} \Big\} dx \\
&+ (\Delta T) \sum_{m=1}^{N_l} \int_0^L \Big\{ S_1^{(m)} \delta u_0^{(m)} + S_2^{(m)} \delta u_1^{(m)} + S_3^{(m)} \delta u_2^{(m)} + S_4^{(m)} \delta u_3^{(m)} + S_5^{(m)} \delta w_0^{(m)} + S_6^{(m)} \delta w_1^{(m)} + S_7^{(m)} \delta w_2^{(m)} + S_8^{(m)} \delta w_3^{(m)} \Big\} dx \\
&+ \sum_{k=1}^{(N_l-1)} \int_0^L \Big[ \left(\delta \lambda_x^{(k)}\right)\left(f_x^{(k)}\right) + \left(\delta \lambda_z^{(k)}\right)\left(f_z^{(k)}\right) + \left(\lambda_x^{(k)}\right)\left(\delta f_x^{(k)}\right) + \left(\lambda_z^{(k)}\right)\left(\delta f_z^{(k)}\right) \Big] dx + (\text{boundary terms}) = 0.
\end{aligned}
\tag{15}
$$

According to Equation (15), the Euler–Lagrange equations of the mixed LW HSDT can be obtained as follows:

$$\delta u_0^{(m)} : \quad -N_x^{(m)},_x + \left(\lambda_x^{(m-1)} - \lambda_x^{(m)}\right) = -(\Delta T)S_1^{(m)}, \tag{16}$$

$$\delta u_1^{(m)} : \quad -M_x^{(m)},_x + N_{xz}^{(m)} + (-h_m/2)\left(\lambda_x^{(m-1)} + \lambda_x^{(m)}\right) = -(\Delta T)S_2^{(m)}, \tag{17}$$

$$\delta u_2^{(m)} : \quad -P_x^{(m)},_x + 2M_{xz}^{(m)} + \left(h_m^2/4\right)\left(\lambda_x^{(m-1)} - \lambda_x^{(m)}\right) = -(\Delta T)S_3^{(m)}, \tag{18}$$

$$\delta u_3^{(m)} : \quad -R_x^{(m)},_x + 3P_{xz}^{(m)} + \left(-h_m^3/8\right)\left(\lambda_x^{(m-1)} + \lambda_x^{(m)}\right) = -(\Delta T)S_4^{(m)}, \tag{19}$$

$$\delta w_0^{(m)} : \quad -N_{xz}^{(m)},_x + \left(\lambda_z^{(m-1)} - \lambda_z^{(m)}\right) = -(\Delta T)S_5^{(m)}, \tag{20}$$

$$\delta w_1^{(m)} : \quad -M_{xz}^{(m)},_x + N_z^{(m)} + (-h_m/2)\left(\lambda_z^{(m-1)} + \lambda_z^{(m)}\right) = -(\Delta T)S_6^{(m)}, \tag{21}$$

$$\delta w_2^{(m)} : \quad -P_{xz}^{(m)},_x + 2M_z^{(m)} + \left(h_m^2/4\right)\left(\lambda_z^{(m-1)} - \lambda_z^{(m)}\right) = -(\Delta T)S_7^{(m)}, \tag{22}$$

$$\delta w_3^{(m)} : \quad -R_{xz}^{(m)},_x + 3P_z^{(m)} + \left(-h_m^3/8\right)\left(\lambda_z^{(m-1)} + \lambda_z^{(m)}\right) = -(\Delta T)S_8^{(m)}, \tag{23}$$

$$\delta \lambda_x^{(k)} : \quad f_x^{(k)} = 0, \tag{24}$$

$$\delta \lambda_z^{(k)} : \quad f_z^{(k)} = 0, \tag{25}$$

where $m = 1, 2, \ldots, N_l$ and $k = 1, 2, \ldots, (N_l - 1)$.

The possible boundary conditions are given as

$$\text{Either } N_x^{(m)} = N_{xt}^{(m)} \text{ or } u_0^{(m)} = \overline{u}_0^{(m)} \tag{26}$$

$$\text{Either } M_x^{(m)} = M_{xt}^{(m)} \text{ or } u_1^{(m)} = \overline{u}_1^{(m)}, \tag{27}$$

$$\text{Either } P_x^{(m)} = P_{xt}^{(m)} \text{ or } u_2^{(m)} = \overline{u}_2^{(m)}, \tag{28}$$

$$\text{Either } R_x^{(m)} = R_{xt}^{(m)} \text{ or } u_3^{(m)} = \overline{u}_3^{(m)}, \tag{29}$$

$$\text{Either } N_{xz}^{(m)} = N_{xzt}^{(m)} \text{ or } w_0^{(m)} = \overline{w}_0^{(m)}, \tag{30}$$

$$\text{Either } M_{xz}^{(m)} = M_{xzt}^{(m)} \text{ or } w_1^{(m)} = \overline{w}_1^{(m)}, \tag{31}$$

$$\text{Either } P_{xz}^{(m)} = P_{xzt}^{(m)} \text{ or } w_2^{(m)} = \overline{w}_2^{(m)}, \tag{32}$$

$$\text{Either } R_{xz}^{(m)} = R_{xzt}^{(m)} \text{ or } w_3^{(m)} = \overline{w}_3^{(m)}, \tag{33}$$

where $\overline{u}_i^{(m)}$ and $\overline{w}_i^{(m)}$ ($i = 0, 1, 2,$ and $3$) are the prescribed displacement components on the edges. The definition of each force resultant component and its relationship with the displacement components are given in Appendix A.

Substituting Equations (A1)–(A11) into Equations (16)–(25), the authors obtain the Euler–Lagrange equations of the current mixed LW HSDBT in terms of all displacement components, and they are given as follows:

$$
\begin{aligned}
\delta u_0^{(m)} : \; & -A_{11}^{(m)} u_0^{(m)},_{xx} - B_{11}^{(m)} u_1^{(m)},_{xx} - D_{11}^{(m)} u_2^{(m)},_{xx} - F_{11}^{(m)} u_3^{(m)},_{xx} - A_{13}^{(m)} w_1^{(m)},_x - 2B_{13}^{(m)} w_2^{(m)},_x \\
& -3D_{13}^{(m)} w_3^{(m)},_x + \left( \lambda_x^{(m-1)} - \lambda_x^{(m)} \right) = -(\Delta T) \left( A_{1\alpha}^{(m)} u_0^{(m)},_{xx} + B_{1\alpha}^{(m)} u_1^{(m)},_{xx} + D_{1\alpha}^{(m)} u_2^{(m)},_{xx} + F_{1\alpha}^{(m)} u_3^{(m)},_{xx} \right),
\end{aligned}
\tag{34}
$$

$$
\begin{aligned}
\delta u_1^{(m)} : \; & -B_{11}^{(m)} u_0^{(m)},_{xx} - D_{11}^{(m)} u_1^{(m)},_{xx} - F_{11}^{(m)} u_2^{(m)},_{xx} - H_{11}^{(m)} u_3^{(m)},_{xx} - B_{13}^{(m)} w_1^{(m)},_x - 2D_{13}^{(m)} w_2^{(m)},_x \\
& -3F_{13}^{(m)} w_3^{(m)},_x + A_{55}^{(m)} \left( w_0^{(m)},_x + u_1^{(m)} \right) + B_{55}^{(m)} \left( w_1^{(m)},_x + 2u_2^{(m)} \right) + D_{55}^{(m)} \left( w_2^{(m)},_x + 3u_3^{(m)} \right) \\
& +F_{55}^{(m)} w_3^{(m)},_x + (-h_m/2) \left( \lambda_x^{(m-1)} + \lambda_x^{(m)} \right) = -(\Delta T) \left( B_{1\alpha}^{(m)} u_0^{(m)},_{xx} + D_{1\alpha}^{(m)} u_1^{(m)},_{xx} + F_{1\alpha}^{(m)} u_2^{(m)},_{xx} + H_{1\alpha}^{(m)} u_3^{(m)},_{xx} \right),
\end{aligned}
\tag{35}
$$

$$
\begin{aligned}
\delta u_2^{(m)} : \; & -D_{11}^{(m)} u_0^{(m)},_{xx} - F_{11}^{(m)} u_1^{(m)},_{xx} - H_{11}^{(m)} u_2^{(m)},_{xx} - J_{11}^{(m)} u_3^{(m)},_{xx} - D_{13}^{(m)} w_1^{(m)},_x - 2F_{13}^{(m)} w_2^{(m)},_x \\
& -3H_{13}^{(m)} w_3^{(m)},_x + 2B_{55}^{(m)} \left( w_0^{(m)},_x + u_1^{(m)} \right) + 2D_{55}^{(m)} \left( w_1^{(m)},_x + 2u_2^{(m)} \right) + 2F_{55}^{(m)} \left( w_2^{(m)},_x + 3u_3^{(m)} \right) \\
& +2H_{55}^{(m)} w_3^{(m)},_x + (h_m^2/4) \left( \lambda_x^{(m-1)} + \lambda_x^{(m)} \right) = -(\Delta T) \left( D_{1\alpha}^{(m)} u_0^{(m)},_{xx} + F_{1\alpha}^{(m)} u_1^{(m)},_{xx} + H_{1\alpha}^{(m)} u_2^{(m)},_{xx} + J_{1\alpha}^{(m)} u_3^{(m)},_{xx} \right),
\end{aligned}
\tag{36}
$$

$$
\begin{aligned}
\delta u_3^{(m)} : \; & -F_{11}^{(m)} u_0^{(m)},_{xx} - H_{11}^{(m)} u_1^{(m)},_{xx} - J_{11}^{(m)} u_2^{(m)},_{xx} - L_{11}^{(m)} u_3^{(m)},_{xx} - F_{13}^{(m)} w_1^{(m)},_x - 2H_{13}^{(m)} w_2^{(m)},_x \\
& -3J_{13}^{(m)} w_3^{(m)},_x + 3D_{55}^{(m)} \left( w_0^{(m)},_x + u_1^{(m)} \right) + 3F_{55}^{(m)} \left( w_1^{(m)},_x + 2u_2^{(m)} \right) + 3H_{55}^{(m)} \left( w_2^{(m)},_x + 3u_3^{(m)} \right) + 3J_{55}^{(m)} w_3^{(m)},_x \\
& +(-h_m^3/8) \left( \lambda_x^{(m-1)} + \lambda_x^{(m)} \right) = -(\Delta T) \left( F_{1\alpha}^{(m)} u_0^{(m)},_{xx} + H_{1\alpha}^{(m)} u_1^{(m)},_{xx} + J_{1\alpha}^{(m)} u_2^{(m)},_{xx} + L_{1\alpha}^{(m)} u_3^{(m)},_{xx} \right),
\end{aligned}
\tag{37}
$$

$$
\begin{aligned}
\delta w_0^{(m)} : \; & -A_{55}^{(m)} \left( w_0^{(m)},_{xx} + u_1^{(m)},_x \right) - B_{55}^{(m)} \left( w_1^{(m)},_{xx} + 2 u_2^{(m)},_x \right) - D_{55}^{(m)} \left( w_2^{(m)},_{xx} + 3 u_3^{(m)},_x \right) - F_{55}^{(m)} w_3^{(m)},_{xx} \\
& + \left( \lambda_z^{(m-1)} - \lambda_z^{(m)} \right) = -(\Delta T) \left( A_{1\alpha}^{(m)} w_0^{(m)},_{xx} + B_{1\alpha}^{(m)} w_1^{(m)},_{xx} + D_{1\alpha}^{(m)} w_2^{(m)},_{xx} + F_{1\alpha}^{(m)} w_3^{(m)},_{xx} \right),
\end{aligned}
\tag{38}
$$

$$
\begin{aligned}
\delta w_1^{(m)} : \; & -B_{55}^{(m)} \left( w_0^{(m)},_{xx} + u_1^{(m)},_x \right) - D_{55}^{(m)} \left( w_1^{(m)},_{xx} + 2 u_2^{(m)},_x \right) - F_{55}^{(m)} \left( w_2^{(m)},_{xx} + 3 u_3^{(m)},_x \right) - H_{55}^{(m)} w_3^{(m)},_{xx} \\
& +A_{13}^{(m)} u_0^{(m)},_x + B_{13}^{(m)} u_1^{(m)},_x + D_{13}^{(m)} u_2^{(m)},_x + F_{13}^{(m)} u_3^{(m)},_x + A_{33}^{(m)} w_1^{(m)} + 2B_{33}^{(m)} w_2^{(m)} + 3D_{33}^{(m)} w_3^{(m)} \\
& +(-h_m/2) \left( \lambda_z^{(m-1)} + \lambda_z^{(m)} \right) = -(\Delta T) \left( B_{1\alpha}^{(m)} w_0^{(m)},_{xx} + D_{1\alpha}^{(m)} w_1^{(m)},_{xx} + F_{1\alpha}^{(m)} w_2^{(m)},_{xx} + H_{1\alpha}^{(m)} w_3^{(m)},_{xx} \right),
\end{aligned}
\tag{39}
$$

$$
\begin{aligned}
\delta w_2^{(m)} : \; & -D_{55}^{(m)} \left( w_0^{(m)},_{xx} + u_1^{(m)},_x \right) - F_{55}^{(m)} \left( w_1^{(m)},_{xx} + 2 u_2^{(m)},_x \right) - H_{55}^{(m)} \left( w_2^{(m)},_{xx} + 3 u_3^{(m)},_x \right) - J_{55}^{(m)} w_3^{(m)},_{xx} \\
& +2B_{13}^{(m)} u_0^{(m)},_x + 2D_{13}^{(m)} u_1^{(m)},_x + 2F_{13}^{(m)} u_2^{(m)},_x + 2H_{13}^{(m)} u_3^{(m)},_x + 2B_{33}^{(m)} w_1^{(m)} + 4D_{33}^{(m)} w_2^{(m)} + 6F_{33}^{(m)} w_3^{(m)} \\
& +(h_m^2/4) \left( \lambda_z^{(m-1)} - \lambda_z^{(m)} \right) = -(\Delta T) \left( D_{1\alpha}^{(m)} w_0^{(m)},_{xx} + F_{1\alpha}^{(m)} w_1^{(m)},_{xx} + H_{1\alpha}^{(m)} w_2^{(m)},_{xx} + J_{1\alpha}^{(m)} w_3^{(m)},_{xx} \right),
\end{aligned}
\tag{40}
$$

$$
\begin{aligned}
\delta w_3^{(m)} : \; & -F_{55}^{(m)} \left( w_0^{(m)},_{xx} + u_1^{(m)},_x \right) - H_{55}^{(m)} \left( w_1^{(m)},_{xx} + 2 u_2^{(m)},_x \right) - J_{55}^{(m)} \left( w_2^{(m)},_{xx} + 3 u_3^{(m)},_x \right) - L_{55}^{(m)} w_3^{(m)},_{xx} \\
& +3D_{13}^{(m)} u_0^{(m)},_x + 3F_{13}^{(m)} u_1^{(m)},_x + 3H_{13}^{(m)} u_2^{(m)},_x + 3J_{13}^{(m)} u_3^{(m)},_x + 3D_{33}^{(m)} w_1^{(m)} + 6F_{33}^{(m)} w_2^{(m)} + 9H_{33}^{(m)} w_3^{(m)} \\
& +(-h_m^3/8) \left( \lambda_z^{(m-1)} + \lambda_z^{(m)} \right) = -(\Delta T) \left( F_{1\alpha}^{(m)} w_0^{(m)},_{xx} + H_{1\alpha}^{(m)} w_1^{(m)},_{xx} + J_{1\alpha}^{(m)} w_2^{(m)},_{xx} + L_{1\alpha}^{(m)} w_3^{(m)},_{xx} \right),
\end{aligned}
\tag{41}
$$

$$
\delta \lambda_x^{(k)} : \; f_x^{(k)} = 0,
\tag{42}
$$

$$
\delta \lambda_z^{(k)} : \; f_z^{(k)} = 0,
\tag{43}
$$

where $m = 1, 2, \dots, N_l$ and $k = 1, 2, \dots, (N_l - 1)$.

The total number of Euler–Lagrange equations, i.e., Equations (34)–(43), is $(10N_l - 2)$, $(8N_l - 2)$, and $(6N_l - 2)$ for the mixed layer-wise third-order (LW3), second-order (LW2), and first-order (LW1) shear deformation theories, respectively, are in terms of the same num-

ber of unknowns as those of the corresponding Euler–Lagrange equations. The strong-form formulations of the mixed LW HSDT are, thus, obtained, including the Euler–Lagrange equations (Equations (34)–(43)) and the possible boundary conditions (Equations (26)–(33)).

### 3.2. Applications

Equations (34)–(43) associated with a set of boundary conditions (i.e., Equations (26)–(33)) can be composed as a well-post boundary value problem, and its Navier-type analytical solutions for the thermal buckling behavior of simply supported, multi-layered FG beams can be obtained using the Fourier series expansion method.

By satisfying the simply supported boundary conditions, various field variables of the *m*th-layer are, thus, expanded as

$$u_i^{(m)} = \sum_{\hat{m}=1}^{\infty} u_{i\hat{m}}^{(m)} \cos(\hat{m}\pi x/L), \tag{44}$$

$$w_i^{(m)} = \sum_{\hat{m}=1}^{\infty} w_{i\hat{m}}^{(m)} \sin(\hat{m}\pi x/L), \tag{45}$$

$$\lambda_x^{(k)} = \sum_{\hat{m}=1}^{\infty} \lambda_{x\hat{m}}^{(k)} \cos(\hat{m}\pi x/L), \tag{46}$$

$$\lambda_z^{(k)} = \sum_{\hat{m}=1}^{\infty} \lambda_{z\hat{m}}^{(k)} \sin(\hat{m}\pi x/L), \tag{47}$$

where the subscript $i = 0, 1, 2,$ and $3$; $m = 1, 2, \ldots, N_l$; $k = 1, 2, \ldots, (N_l - 1)$.

Substituting Equations (44)–(47) in the Euler–Lagrange Equations (i.e., Equations (34)–(43)), the authors obtain

$$\delta u_0^{(m)} : \overline{m}^2 A_{11}^{(m)} u_{0\hat{m}}^{(m)} + \overline{m}^2 B_{11}^{(m)} u_{1\hat{m}}^{(m)} + \overline{m}^2 D_{11}^{(m)} u_{2\hat{m}}^{(m)} + \overline{m}^2 F_{11}^{(m)} u_{3\hat{m}}^{(m)} - \overline{m} A_{13}^{(m)} w_{1\hat{m}}^{(m)} - 2\overline{m} B_{13}^{(m)} w_{2\hat{m}}^{(m)}$$
$$-3\overline{m} D_{13}^{(m)} w_{3\hat{m}}^{(m)} + \left( \lambda_{x\hat{m}}^{(m-1)} - \lambda_{x\hat{m}}^{(m)} \right) = (\Delta T) \left( \overline{m}^2 A_{1\alpha}^{(m)} u_{0\hat{m}}^{(m)} + \overline{m}^2 B_{1\alpha}^{(m)} u_{1\hat{m}}^{(m)} + \overline{m}^2 D_{1\alpha}^{(m)} u_{2\hat{m}}^{(m)} + \overline{m}^2 F_{1\alpha}^{(m)} u_{3\hat{m}}^{(m)} \right), \tag{48}$$

$$\delta u_1^{(m)} : \overline{m}^2 B_{11}^{(m)} u_{0\hat{m}}^{(m)} + \left( \overline{m}^2 D_{11}^{(m)} + A_{55}^{(m)} \right) u_{1\hat{m}}^{(m)} + \left( \overline{m}^2 F_{11}^{(m)} + 2B_{55}^{(m)} \right) u_{2\hat{m}}^{(m)} + \left( \overline{m}^2 H_{11}^{(m)} + 3D_{55}^{(m)} \right) u_{3\hat{m}}^{(m)}$$
$$+ \overline{m} A_{55}^{(m)} w_{0\hat{m}}^{(m)} + \left( -\overline{m} B_{13}^{(m)} + \overline{m} B_{55}^{(m)} \right) w_{1\hat{m}}^{(m)} + \left( -2\overline{m} D_{13}^{(m)} + \overline{m} D_{55}^{(m)} \right) w_{2\hat{m}}^{(m)} + \left( -3\overline{m} F_{13}^{(m)} + \overline{m} F_{55}^{(m)} \right) w_{3\hat{m}}^{(m)}$$
$$+ (-h_m/2) \left( \lambda_{x\hat{m}}^{(m-1)} + \lambda_{x\hat{m}}^{(m)} \right) = (\Delta T) \left( \overline{m}^2 B_{1\alpha}^{(m)} u_{0\hat{m}}^{(m)} + \overline{m}^2 D_{1\alpha}^{(m)} u_{1\hat{m}}^{(m)} + \overline{m}^2 F_{1\alpha}^{(m)} u_{2\hat{m}}^{(m)} + \overline{m}^2 H_{1\alpha}^{(m)} u_{3\hat{m}}^{(m)} \right), \tag{49}$$

$$\delta u_2^{(m)} : \overline{m}^2 D_{11}^{(m)} u_{0\hat{m}}^{(m)} + \left( \overline{m}^2 F_{11}^{(m)} + 2B_{55}^{(m)} \right) u_{1\hat{m}}^{(m)} + \left( \overline{m}^2 H_{11}^{(m)} + 4D_{55}^{(m)} \right) u_{2\hat{m}}^{(m)} + \left( \overline{m}^2 J_{11}^{(m)} + 6F_{55}^{(m)} \right) u_{3\hat{m}}^{(m)}$$
$$+ 2\overline{m} B_{55}^{(m)} w_{0\hat{m}}^{(m)} + \left( -\overline{m} D_{13}^{(m)} + 2\overline{m} D_{55}^{(m)} \right) w_{1\hat{m}}^{(m)} + \left( -2\overline{m} F_{13}^{(m)} + 2\overline{m} F_{55}^{(m)} \right) w_{2\hat{m}}^{(m)} + \left( -3\overline{m} H_{13}^{(m)} + 2\overline{m} H_{55}^{(m)} \right) w_{3\hat{m}}^{(m)}$$
$$+ (h_m^2/4) \left( \lambda_{x\hat{m}}^{(m-1)} - \lambda_{x\hat{m}}^{(m)} \right) = (\Delta T) \left( \overline{m}^2 D_{1\alpha}^{(m)} u_{0\hat{m}}^{(m)} + \overline{m}^2 F_{1\alpha}^{(m)} u_{1\hat{m}}^{(m)} + \overline{m}^2 H_{1\alpha}^{(m)} u_{2\hat{m}}^{(m)} + \overline{m}^2 J_{1\alpha}^{(m)} u_{3\hat{m}}^{(m)} \right), \tag{50}$$

$$\delta u_3^{(m)} : \overline{m}^2 F_{11}^{(m)} u_{0\hat{m}}^{(m)} + \left( \overline{m}^2 H_{11}^{(m)} + 3D_{55}^{(m)} \right) u_{1\hat{m}}^{(m)} + \left( \overline{m}^2 J_{11}^{(m)} + 6F_{55}^{(m)} \right) u_{2\hat{m}}^{(m)} + \left( \overline{m}^2 L_{11}^{(m)} + 9H_{55}^{(m)} \right) u_{3\hat{m}}^{(m)}$$
$$+ 3\overline{m} D_{55}^{(m)} w_{0\hat{m}}^{(m)} + \left( -\overline{m} F_{13}^{(m)} + 3\overline{m} F_{55}^{(m)} \right) w_{1\hat{m}}^{(m)} + \left( -2\overline{m} H_{13}^{(m)} + 3\overline{m} H_{55}^{(m)} \right) w_{2\hat{m}}^{(m)} + \left( -3\overline{m} J_{13}^{(m)} + 3\overline{m} J_{55}^{(m)} \right) w_{3\hat{m}}^{(m)}$$
$$+ (-h_m^3/8) \left( \lambda_{x\hat{m}}^{(m-1)} + \lambda_{x\hat{m}}^{(m)} \right) = (\Delta T) \left( \overline{m}^2 F_{1\alpha}^{(m)} u_{0\hat{m}}^{(m)} + H^2 F_{1\alpha}^{(m)} u_{1\hat{m}}^{(m)} + \overline{m}^2 J_{1\alpha}^{(m)} u_{2\hat{m}}^{(m)} + \overline{m}^2 L_{1\alpha}^{(m)} u_{3\hat{m}}^{(m)} \right), \tag{51}$$

$$\delta w_0^{(m)} : \overline{m} A_{55}^{(m)} u_{1\hat{m}}^{(m)} + 2\overline{m} B_{55}^{(m)} u_{2\hat{m}}^{(m)} + 3\overline{m} D_{55}^{(m)} u_{3\hat{m}}^{(m)} + \overline{m}^2 A_{55}^{(m)} w_{0\hat{m}}^{(m)} + \overline{m}^2 B_{55}^{(m)} w_{1\hat{m}}^{(m)} + \overline{m}^2 D_{55}^{(m)} w_{2\hat{m}}^{(m)}$$
$$+ \overline{m}^2 F_{55}^{(m)} w_{3\hat{m}}^{(m)} + \left( \lambda_{z\hat{m}}^{(m-1)} - \lambda_{z\hat{m}}^{(m)} \right) = (\Delta T) \left( \overline{m}^2 A_{1\alpha}^{(m)} w_{0\hat{m}}^{(m)} + \overline{m}^2 B_{1\alpha}^{(m)} w_{1\hat{m}}^{(m)} + \overline{m}^2 D_{1\alpha}^{(m)} w_{2\hat{m}}^{(m)} + \overline{m}^2 F_{1\alpha}^{(m)} w_{3\hat{m}}^{(m)} \right), \tag{52}$$

$$\delta w_1^{(m)} : -\overline{m} A_{13}^{(m)} u_{0\hat{m}}^{(m)} + \left( -\overline{m} B_{13}^{(m)} + \overline{m} B_{55}^{(m)} \right) u_{1\hat{m}}^{(m)} + \left( -\overline{m} D_{13}^{(m)} + 2\overline{m} D_{55}^{(m)} \right) u_{2\hat{m}}^{(m)} + \left( -\overline{m} F_{13}^{(m)} + 3\overline{m} F_{55}^{(m)} \right) u_{3\hat{m}}^{(m)}$$
$$+ \overline{m}^2 B_{55}^{(m)} w_{0\hat{m}}^{(m)} + \left( A_{13}^{(m)} + \overline{m}^2 D_{55}^{(m)} \right) w_{1\hat{m}}^{(m)} + \left( 2B_{33}^{(m)} + \overline{m}^2 F_{55}^{(m)} \right) w_{2\hat{m}}^{(m)} + \left( 3D_{33}^{(m)} + \overline{m}^2 H_{55}^{(m)} \right) w_{3\hat{m}}^{(m)}$$
$$+ (-h_m/2) \left( \lambda_{x\hat{m}}^{(m-1)} + \lambda_{x\hat{m}}^{(m)} \right) = (\Delta T) \left( \overline{m}^2 B_{1\alpha}^{(m)} w_{0\hat{m}}^{(m)} + \overline{m}^2 D_{1\alpha}^{(m)} w_{1\hat{m}}^{(m)} + \overline{m}^2 F_{1\alpha}^{(m)} w_{2\hat{m}}^{(m)} + \overline{m}^2 H_{1\alpha}^{(m)} w_{3\hat{m}}^{(m)} \right), \tag{53}$$

$$\delta w_2^{(m)} : \quad -2\overline{m}B_{13}^{(m)} u_{0\hat{m}}^{(m)} + \left(-2\overline{m}D_{13}^{(m)} + \overline{m}D_{55}^{(m)}\right) u_{1\hat{m}}^{(m)} + \left(-2\overline{m}F_{13}^{(m)} + 2\overline{m}F_{55}^{(m)}\right) u_{2\hat{m}}^{(m)} + \left(-2\overline{m}H_{13}^{(m)} + 3\overline{m}H_{55}^{(m)}\right) u_{3\hat{m}}^{(m)}$$

$$+\overline{m}^2 D_{55}^{(m)} w_{0\hat{m}}^{(m)} + \left(2B_{33}^{(m)} + \overline{m}^2 F_{55}^{(m)}\right) w_{1\hat{m}}^{(m)} + \left(4D_{33}^{(m)} + \overline{m}^2 H_{55}^{(m)}\right) w_{2\hat{m}}^{(m)} + \left(6F_{33}^{(m)} + \overline{m}^2 J_{55}^{(m)}\right) w_{3\hat{m}}^{(m)} \tag{54}$$

$$+\left(h_m^2/4\right)\left(\lambda_{z\hat{m}}^{(m-1)} - \lambda_{z\hat{m}}^{(m)}\right) = (\Delta T)\left(\overline{m}^2 D_{1\alpha}^{(m)} w_{0\hat{m}}^{(m)} + \overline{m}^2 F_{1\alpha}^{(m)} w_{1\hat{m}}^{(m)} + \overline{m}^2 H_{1\alpha}^{(m)} w_{2\hat{m}}^{(m)} + \overline{m}^2 J_{1\alpha}^{(m)} w_{3\hat{m}}^{(m)}\right),$$

$$\delta w_3^{(m)} : \quad -3\overline{m}D_{13}^{(m)} u_{0\hat{m}}^{(m)} + \left(-3\overline{m}F_{13}^{(m)} + \overline{m}F_{55}^{(m)}\right) u_{1\hat{m}}^{(m)} + \left(-3\overline{m}H_{13}^{(m)} + 2\overline{m}H_{55}^{(m)}\right) u_{2\hat{m}}^{(m)} + \left(-3\overline{m}J_{13}^{(m)} + 3\overline{m}J_{55}^{(m)}\right) u_{3\hat{m}}^{(m)}$$

$$+\overline{m}^2 F_{55}^{(m)} w_{0\hat{m}}^{(m)} + \left(3D_{33}^{(m)} + \overline{m}^2 H_{55}^{(m)}\right) w_{1\hat{m}}^{(m)} + \left(6F_{33}^{(m)} + \overline{m}^2 J_{55}^{(m)}\right) w_{2\hat{m}}^{(m)} + \left(9H_{33}^{(m)} + \overline{m}^2 L_{55}^{(m)}\right) w_{3\hat{m}}^{(m)} \tag{55}$$

$$+\left(-h_m^3/8\right)\left(\lambda_{z\hat{m}}^{(m-1)} + \lambda_{z\hat{m}}^{(m)}\right) = (\Delta T)\left(\overline{m}^2 F_{1\alpha}^{(m)} w_{0\hat{m}}^{(m)} + \overline{m}^2 H_{1\alpha}^{(m)} w_{1\hat{m}}^{(m)} + \overline{m}^2 J_{1\alpha}^{(m)} w_{2\hat{m}}^{(m)} + \overline{m}^2 L_{1\alpha}^{(m)} w_{3\hat{m}}^{(m)}\right),$$

$$\delta\lambda_x^{(k)} : \quad \left[u_{0\hat{m}}^{(k+1)} - (h_{k+1}/2) u_{1\hat{m}}^{(k+1)} + \left(h_{k+1}^2/4\right) u_{2\hat{m}}^{(k+1)} - \left(h_{k+1}^3/8\right) u_{3\hat{m}}^{(k+1)}\right]$$

$$-\left[u_{0\hat{m}}^{(k)} + (h_k/2) u_{1\hat{m}}^{(k)} + \left(h_k^2/4\right) u_{2\hat{m}}^{(k)} + \left(h_k^3/8\right) u_{3\hat{m}}^{(k)}\right] = 0, \tag{56}$$

$$\delta\lambda_z^{(k)} : \quad \left[w_{0\hat{m}}^{(k+1)} - (h_{k+1}/2) w_{1\hat{m}}^{(k+1)} + \left(h_{k+1}^2/4\right) w_{2\hat{m}}^{(k+1)} - \left(h_{k+1}^3/8\right) w_{3\hat{m}}^{(k+1)}\right]$$

$$-\left[w_{0\hat{m}}^{(k)} + (h_k/2) w_{1\hat{m}}^{(k)} + \left(h_k^2/4\right) w_{2\hat{m}}^{(k)} + \left(h_k^3/8\right) w_{3\hat{m}}^{(k)}\right] = 0, \tag{57}$$

where $\overline{m} = \hat{m}\,\pi/L$, $m = 1, 2, \ldots, N_l$, and $k = 1, 2, \ldots, (N_l - 1)$.

Equations (48)–(57) are a system of simultaneously linear algebraic equations representing a standard eigen-valued problem, in which the eigenvalues are the critical temperature changes, which can be obtained by letting the determinant of the coefficient matrix of these simultaneously linear algebraic equations be zero. Once the eigenvalues are obtained, their corresponding eigen-vectors (i.e., buckling mode shapes) can also be determined.

## 4. Optimal Design

### 4.1. Statement of the Optimization Problem

In this work, the authors consider the multi-objective optimization of the volume fractions of the constituents of FG beams subjected to a uniform temperature change in order to maximize the critical temperature change parameter of the FG beam and to minimize its total mass. The FG beam is considered to be formed by mixing a metal material and a ceramic material according to a three-parameter power-law function of the volume fractions of the constituents through the thickness coordinate of the FG beam, which is given as follows:

The three-parameter power-law function,

$$\Gamma_c = \left[\left(\frac{1}{2} + \frac{z}{h}\right) + \kappa_b \left(\frac{1}{2} - \frac{z}{h}\right)^{\kappa_c}\right]^{\kappa_p} \text{ and } \Gamma_c + \Gamma_m = 1, \tag{58}$$

where the symbols $\kappa_b$, $\kappa_c$, and $\kappa_p$ denote the material-property gradient indices. The material at the top surface of the FG beam (i.e., $z = h/2$) is ceramic rich, and when $\kappa_b = 0$, the material at the bottom surface of the FG beam (i.e., $z = -h/2$) is metal rich. Variations in the through-thickness distributions of the volume fraction given in Equation (58) with varying one of $\kappa_b$, $\kappa_c$, and $\kappa_p$ and holding the other two indices constants are shown in Figure 2.

In addition, when $\kappa_b = 0$, the three-parameter power-law function is reduced to the one-parameter power-law function, which is commonly used in the literature and is given as follows:

One-parameter power-law function,

$$\Gamma_c = \left(\frac{1}{2} + \frac{z}{h}\right)^{\kappa_p} \text{ and } \Gamma_c + \Gamma_m = 1, \tag{59}$$

which is used in the thermal buckling cases in this work for a comparative study. Variations in the through-thickness distributions of the volume fraction given in Equation (59) with varying the $\kappa_p$ value are shown in Figure 3.

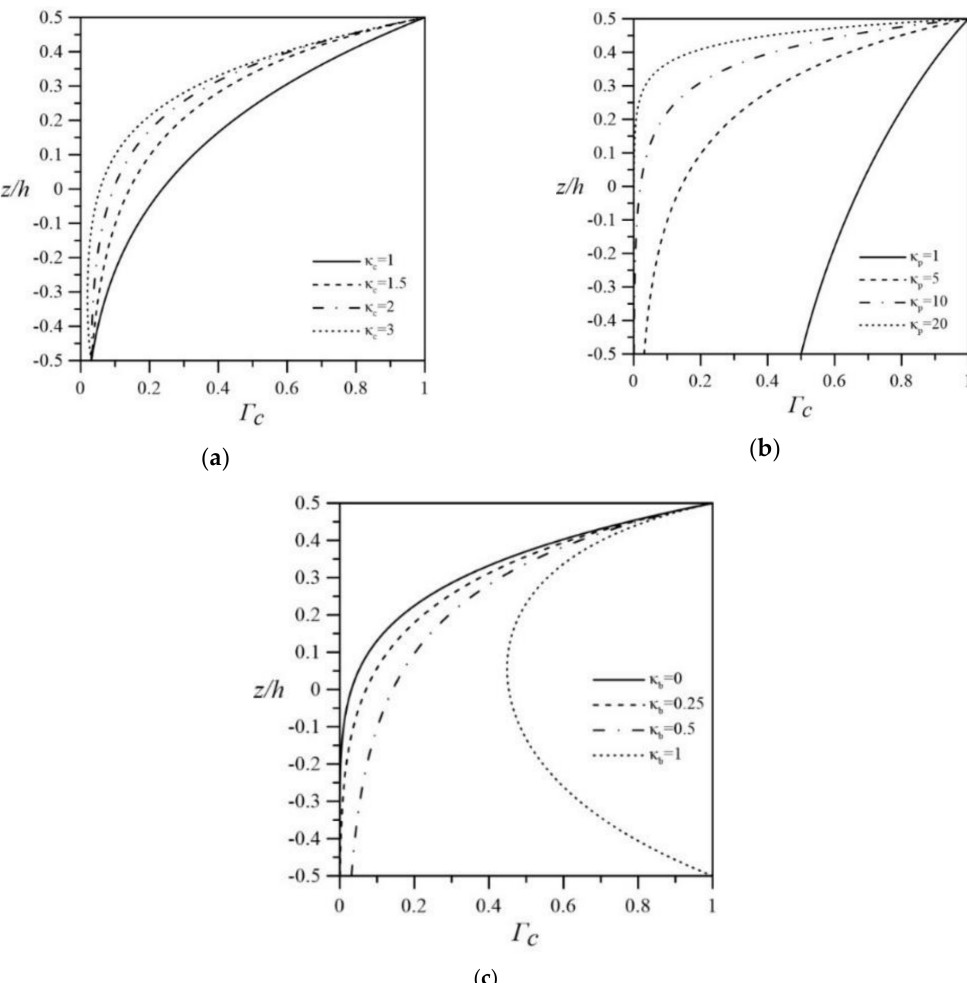

**Figure 2.** Through-thickness distributions of the volume fractions of the ceramic material for a three-parameter power-law material model when (**a**) $\kappa_b = 0.5$, $\kappa_p = 5$, and $\kappa_c = 1$, 1.5, 2, and 3; (**b**) $\kappa_b = 0.5$, $\kappa_c = 1.5$, and $\kappa_p = 1$, 5, 10, and 20; and (**c**) $\kappa_p = 5$, $\kappa_c = 1.5$, and $\kappa_b = 0$, 0.025, 0.5, and 1.

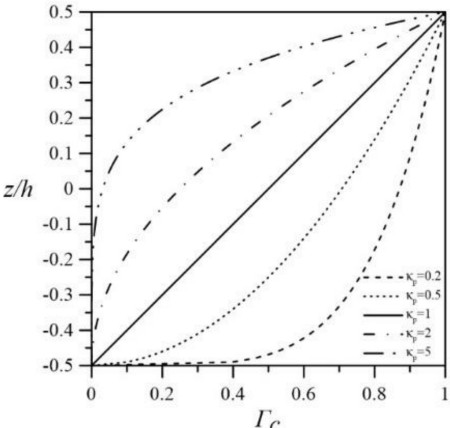

**Figure 3.** Through-thickness distributions of volume fractions of the ceramic material for a one-parameter power-law material model when $\kappa_p = 0.2$, 0.5, 1, 2, and 5.

In the optimal design, a mass ratio $R_m$ is defined as follows:

$$R_m = \left(\hat{\rho}_f - \hat{\rho}_c\right) / \left(\hat{\rho}_m - \hat{\rho}_c\right), \tag{60}$$

where $\hat{\rho}_f$, $\hat{\rho}_c$, and $\hat{\rho}_m$ are the total mass per unit area in *x-y* plane of the FG beam, the homogeneous ceramic material, and the homogeneous metal material, respectively. $\hat{\rho}_k = \int_{-h/2}^{h/2} \rho_k(z)\, dz$, in which $k = f, c,$ or $m$.

The critical temperature parameter $\Delta \hat{T}_{cr}$ is defined as follows:

$$\Delta \hat{T}_{cr} = \Delta T_{cr}\, \alpha_c\, (L/h)^2, \tag{61}$$

where $\alpha_c$ denotes the thermal expansion coefficient of a reference ceramic material.

The objective functions are defined as follows:

$$\text{Objective function 1}: \ F_1 = R_m, \tag{62}$$

$$\text{Objective function 2}: \ F_2 = 1 - \{ [(\Delta T_{cr}) - (\Delta T_{cr})_c]/[(\Delta T_{cr})_m - (\Delta T_{cr})_c] \}, \tag{63}$$

where the ranges of $F_1$ and $F_2$ are $0 \le F_1(\text{or } F_2) \le 1$.

Because non-dominated sorting is used in the current GA, the values of these objective functions given in Equations (62) and (63) are, thus, used to classify each design into its corresponding non-dominated front, which is also the assigned fitness value used for the sorting process. Minimization of the fitness function can be accomplished when the critical temperature change parameters of the FG beam are obtained. The detailed process of the current non-dominated sorting-based GA is described in the following sections.

### 4.2. The Non-Dominated Sorting-Based GA

In this work, the non-dominated sorting-based GA [69,70] is used as an optimal technique, the usual form of which is described as follows: The GA starts with an initial set of random solutions, which are called a population. Each individual in the population is called a chromosome, which represents a solution to the problem at hand. The chromosomes are made of discrete units called genes (or design variables). Each gene controls one or more features of the chromosome. The chromosomes evolve through successive iterations, called generations. Based on some measures of fitness, the chromosomes during each generation are evaluated. To create the next generation, new chromosomes in the next generation, called offspring, are formed using two operators, which are the crossover and mutation operators. In the former, some portions of two chromosomes in the current generation are merged together, and in the latter, some portions of chromosomes in the current generation are modified. Thus, the crossover operator leads the population to converge by means of making the chromosome in the population alike, and the mutation operator assists the search escaping from local optima by reintroducing genetic diversity back into the population. A new generation is formed by selecting and rejecting some of the parents and offspring according to their fitness values so as to keep the population size of each generation constant. After several generations, the GA converges to the best chromosome, representing the optimum solution to the problem considered. A flow chart of the non-dominated sorting-based GA is shown in Figure 4, and its related process is described as follows:

(a)    Initial population

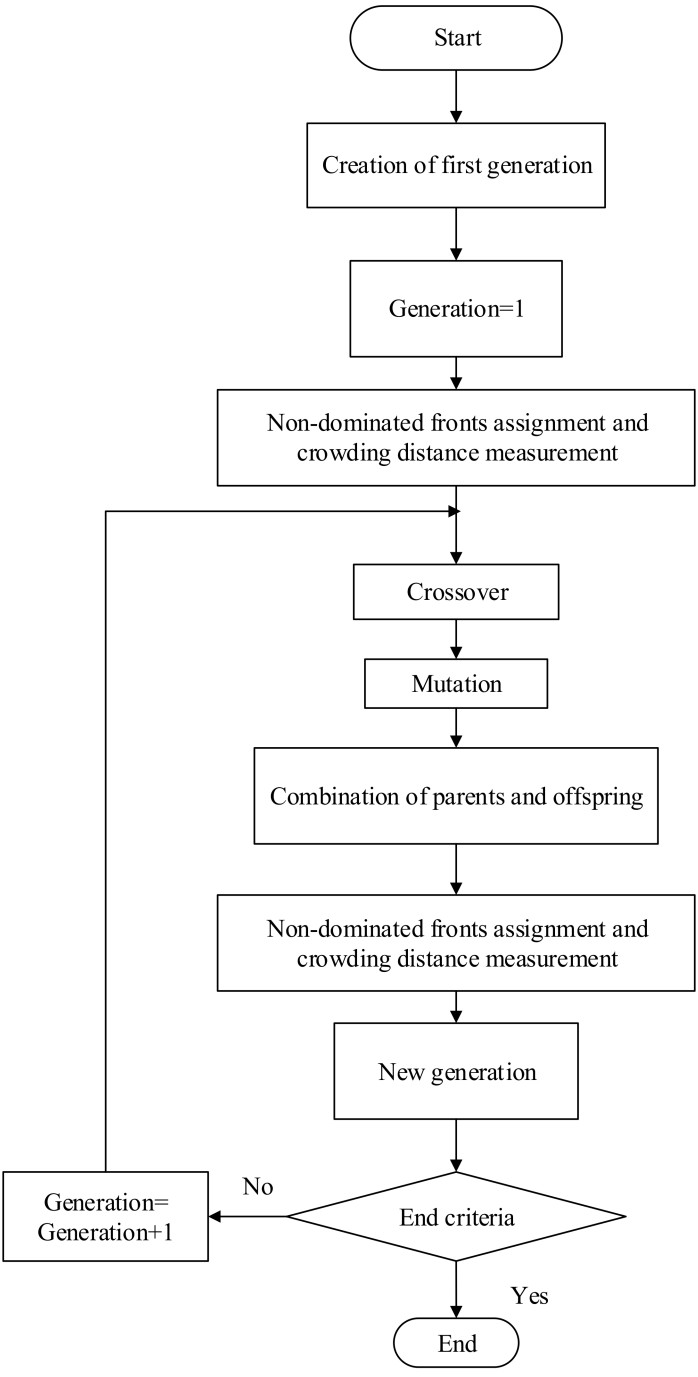

**Figure 4.** Flow chart of the non-dominated sorting-based genetic algorithm (GA).

The first population consists of chromosomes (i.e., solutions or designs), which is taken to be 200 in this work. Each chromosome is formed by several genes (i.e., design variables), each of which is represented by a randomly generated real number with four decimal places in a specific range.

(b)    Fitness value

As mentioned before, there are two objective functions considered in this work, the critical temperature change parameter of the FG beam and its total mass, the specific forms of which (i.e., F1 and F2) are, thus, defined as the fitness values and are used for the sorting process in the current GA.

(c)    Selection

In this work, the selection process consists of two approaches, the Pareto-ranking approach [55] and the crowding distance approach [69], which are described as follows:

(c-1) Pareto-ranking approach

The Pareto-ranking approach uses the Pareto dominance concept to evaluate fitness or assign selection probability to solutions. Each solution in the population of the parents is ranked according to a dominance rule, and then each solution is assigned a fitness value based on its rank in the population, rather than its actual objective function value. A dominance rule proposed by Goldberg [55] is adopted and is described as follows:

*Step* 1: Set $i = 1$ and $(n_p)_0 =$ the number of original solutions in the population.

*Step* 2: Identify non-dominated solutions $\mathbf{x}_j$ ($j = 1 \sim n_{ndfi}$) in $(n_p)_i$, and assign them set to $NDF_i$, in which $NDF_i$ denotes the $i$th non-dominated front, and the number of solutions in $NDF_i$ is $n_{ndfi}$.

*Step* 3: Set $i = i + 1$, and then $(n_p)_{i+1} = (n_p)_{i-1} - (n_p)_i$. If $(n_p)_{i+1} = 0$, go to Step 4; otherwise, go to Step 2.

*Step* 4: For every solution $\mathbf{x}_k, k = 1 \sim (n_p)_0$ at generation $t$, assign rank $r(\mathbf{x}_k, t) = i$ if $\mathbf{x}_k \in NDF_i$.

In the above process, a solution $A$ dominates the other solution $B$, which means both $F_1(\mathbf{x}_A) > F_1(\mathbf{x}_B)$ and $F_2(\mathbf{x}_A) > F_2(\mathbf{x}_B)$ when two objective functions are considered. Because all objective functions are to be minimized, a lower rank corresponds to a better solution, such that the population will be classified as several non-dominated fronts, where the first non-dominated front is also called the Pareto fronts of this population.

(c-2) Crowding distance approach

A crowding distance approach as proposed by Deb [69] is adopted in this work, which is aimed toward obtaining a uniform spread of solutions along the best-known Pareto front. The approach is described as follows:

*Step* 1: Rank the population and identify Pareto fronts as $PF_1, PF_2, \ldots, PF_R$, using Equations (62) and (63). For each non-dominated front, repeat Steps 2 and 3.

*Step* 2: For each objective function $F_k$, sort the solutions in $PF_j$ in the ascending order. If the number of solutions in $PF_j$ is $l$, then the crowding distance of the $i$th solution is measured as

$$cd_k(\mathbf{x}_i) = [F_k(\mathbf{x}_{i+1}) - F_k(\mathbf{x}_{i-1})] / \left[ F_k^{\max} - F_k^{\min} \right] \tag{64}$$

where $i = 2, 3, \ldots, (l-1)$, and $cd_k(\mathbf{x}_1) = cd_k(\mathbf{x}_l) = \infty$.

*Step* 3: To obtain the total crowding distance $cd(\mathbf{x}_i) = \sum_{k=1}^{2} cd_k(\mathbf{x}_i)$.

The selection rule is based on the rank assigned to the solutions, where a lower rank corresponds to a better solution. When the solutions are in the same rank, a greater crowding distance corresponds to a better solution.

(d) Crossover

Any two solutions in the population are randomly paired for mating, in which a simulated binary crossover operator suggested by Deb and Goyal [78] is adopted in this work and is given as follows:

$$(x_j)_1^{t+1} = 0.5 \left[ (1 + \gamma_j)(x_j)_1^t + (1 - \gamma_j)(x_j)_2^t \right], \tag{65}$$

$$(x_j)_2^{t+1} = 0.5 \left[ (1 - \gamma_j)(x_j)_1^t + (1 + \gamma_j)(x_j)_2^t \right], \tag{66}$$

where $\gamma_j = (2u_j)^{1/(\eta_j+1)}$ when $u_j \leq 0.5$, and $\gamma_j = \{1/[2(1 - u_j)]\}^{1/(\eta_j+1)}$ when $u_j \geq 0.5$, in which $u_i$ denotes a randomly generated real number between 0 and 1 and $\eta > 0$ is a user-defined index parameter, the value of which is suggested to be $\eta_j = 1$.

(e) Mutation

Mutation occurs in each solution in the population when the randomly generated number between 0 and 1 for each solution is less than 0.1. A polynomial mutation operator suggested by Deb and Goyal [78] is adopted in this work and is given as follows:

$$(x_i)^{t+1} = (x_i)^t + \beta_i, \tag{67}$$

where $\beta_i = (2u_i)^{1/(\eta_i+1)} - 1$ when $u_i \le 0.5$ and $\beta_i = 1 - [2(1 - u_i)]^{1/(\eta_i+1)}$ when $u_i \ge 0.5$, in which $u_i$ denotes a randomly generated real number between 0 and 1 and $\eta_i$ is a user-defined index parameter, for which the value is suggested to be $\eta_i = 20$.

(f) Elitist

Performing the above operators, including the non-dominated sorting, crowding distance, crossover, and mutation approaches, we can obtain the new generation (i.e., offspring) with the same number of solutions (i.e., $n_p$) as those in the original generation (i.e., parents). The elitist strategy in this work is as follows:

*Step* 1: Combine the solutions of parents and offspring to form a population with $2\,n_p$ solutions.

*Step* 2: Perform the non-dominated sorting and crowding distance approaches to each solution of the population generated in Step 1. Then, select the first $n_p$ solutions with smaller fitness values in the population to form the offspring.

(g) End criteria

The end criteria of the current non-dominated sorting-based GA is given as

*Criteria*: when the number of generations reaches an assigned number, which is 100 in this work.

## 5. Illustrative Examples

### 5.1. Thermal Buckling Analysis of Laminated Composite Beams and FG Beams

Because no benchmark solutions for the critical temperature changes for a simply-supported FG beam can be found in the literature, for comparison purposes, the current mixed LW HSDT is applied to the thermal buckling analysis of simply-supported, cross-ply laminated beams under a uniform temperature, in which the material properties are regarded as a layer-wise constant distribution varying through the thickness coordinate. The current solutions for critical temperature change parameters are compared with those obtained using the EBT [79], FSDT [79], TSDT [79], HSDT [80], and HRSDT [22]. The material properties and geometric parameters are given as $L/h = 10$ and $50$, $E_1/E_2 = 10$, $20$, and $30$, $G_{12}/E_2 = G_{13}/E_2 = 0.6$, $v_{12} = v_{13} = 0.25$, $\alpha_2/\alpha_1 = 3$, in which the subscript 1 represents the material properties parallel to the reinforced fiber direction, and the subscripts 2 and 3 represent material properties perpendicular to the reinforced fiber direction. A dimensionless temperature change parameter is defined as $\Delta \overline{T}_{cr} = (\Delta T_{cr})\,\alpha_1 (L/h)^2$.

Tables 1 and 2 show the convergent study for critical temperature change parameter solutions of $[0°/90°/0°]$ laminated beams subjected to a uniform temperature change. It can be seen in Tables 1 and 2 that the current LW1, LW2, and LW3 solutions converge rapidly. For a moderately thick beam ($L/h = 10$), the convergent solutions are yielded when the total number of layers are taken to be 9, 6, and 3 for the LW1, LW2, and LW3 theories, respectively, and these convergent solutions closely agree with one another. The convergence rate increases when the beam becomes thinner. The current solutions in Tables 1 and 2 with a superscript $a$ indicate that the von Kármán GNS, rather than the full GNS, is considered. The results show that the deviations between the solutions considering the von Kármán GNS and the full GNS are less than 1%, which is very minor, such that the von Kármán GNS is suitable for the current analysis of moderately thick laminated composite beams. The current solutions in Tables 1 and 2 with a superscript $b$ indicate that a narrow beam is considered, in which Poisson's ratios are taken to be zeroes in the stiffness coefficients $Q_{ij}$ ($i, j = 1$–$3$). The results show that the deviations between the solutions considering and those not-considering the narrow beam effects are significant, where the

deviations as high as approximately to 5.5% for both a moderately thick ($L/h = 10$) and a thin LC beam ($L/h = 50$). Tables 1 and 2 also show that the current convergent solutions considering the narrow beam effects closely agree with the HSRDT [22] and TSDT [79] solutions, and those not considering the narrow beam effects closely agree with the HSDT solutions [80]. In addition, the EBT solutions for the cases of $L/h = 50$ are the same as those for the cases of $L/h = 10$. In the former, the relative error between the EBT solutions and the current LW HSDT solutions is 1.2%, while it is 31.2% for the latter, such that the EBT is suitable for the current analysis of very thin laminated composite beams only.

**Table 1.** Convergent study of critical temperature parameter solutions of simply supported, laminated ($0°/90°/0°$) beams under a uniform temperature change ($\alpha_2/\alpha_1 = 3$, $E_1/E_2 = 20$).

| $L/h$ | Theories | $\overline{\Delta T}_{cr}$ |
|---|---|---|
| 10 | Current LW1 ($n_l = 1$) | 0.8315 |
| | Current LW1 ($n_l = 3$) | 0.8123 |
| | Current LW1 ($n_l = 6$) | 0.7995 |
| | Current LW1 ($n_l = 9$) | 0.7970 |
| | Current LW2 ($n_l = 1$) | 0.8306 |
| | Current LW2 ($n_l = 3$) | 0.7953 |
| | Current LW2 ($n_l = 6$) | 0.7950 |
| | Current LW3 ($n_l = 1$) | 0.8010 |
| | Current LW3 ($n_l = 3$) | 0.7950 |
| | Current LW3 ($n_l = 6$) | 0.7950 |
| | [a] Current LW3 ($n_l = 6$) | 0.8001 |
| | [b] Current LW3 ($n_l = 6$) | 0.8383 |
| | [a, b] Current LW3 ($n_l = 6$) | 0.8438 |
| | HSDT [80] | 0.78912 |
| | HRSDT [22] | 0.8230 |
| | TSDT [79] | 0.8229 |
| | FSDT [79] | 0.8281 |
| | EBT [79] | 1.1072 |
| 50 | Current LW1 ($n_l = 1$) | 1.0411 |
| | Current LW1 ($n_l = 3$) | 1.0370 |
| | Current LW1 ($n_l = 6$) | 1.0359 |
| | Current LW1 ($n_l = 9$) | 1.0357 |
| | Current LW2 ($n_l = 1$) | 1.0392 |
| | Current LW2 ($n_l = 3$) | 1.0355 |
| | Current LW2 ($n_l = 6$) | 1.0355 |
| | Current LW3 ($n_l = 1$) | 1.0373 |
| | Current LW3 ($n_l = 3$) | 1.0355 |
| | Current LW3 ($n_l = 6$) | 1.0355 |
| | [a] Current LW3 ($n_l = 6$) | 1.0360 |
| | [b] Current LW3 ($n_l = 6$) | 1.0931 |
| | [a, b] Current LW3 ($n_l = 6$) | 1.0936 |
| | HSDT [80] | 1.04656 |
| | HRSDT [22] | 1.0921 |
| | TSDT [79] | 1.0921 |
| | FSDT [79] | 1.0925 |
| | EBT [79] | 1.1072 |

[a] von Karman geometric nonlinear terms are considered. [b] The Poisson ratio is considered to be zero in $Q_{ij}$ ($i = 1$ and 3).

The thermal buckling problems of simply-supported FG beams with either the TD or the TI material properties under a uniform temperature change are considered in Table 3. A flow chart for the solution process to determine the critical temperature change parameters of the FG beam is given in Figure 5 where the TD material properties are considered, in which the golden section method is used. The through-thickness distributions of the material properties of the FG beam are assumed to obey a single-parameter power-law function according to the volume fractions of the constituents, as shown in Equation (57).

The effective material properties are estimated using either the rule of mixtures or the Mori–Tanaka scheme, which are given in Equations (1) and (2), respectively. Again, it can be seen in Table 3 that the current LW HSDT solutions converge rapidly. The solutions for the critical temperature change parameters obtained using the Mori–Tanaka scheme are approximately no more than 2% greater than those obtained using the rule of mixtures. The effects of different micromechanics models on the critical temperature change parameters of the FG beam are minor.

**Table 2.** Convergent study of critical temperature parameter solutions of simply supported, laminated $(0°/90°/0°)$ beams under a uniform temperature change $(\alpha_2/\alpha_1 = 3, L/h = 10)$.

| $E_1/E_2$ | Theories | $\overline{\Delta T_{cr}}$ |
|---|---|---|
| 10 | Current LW1 ($n_l = 1$) | 0.8335 |
| | Current LW1 ($n_l = 3$) | 0.8162 |
| | Current LW1 ($n_l = 6$) | 0.8088 |
| | Current LW1 ($n_l = 9$) | 0.8073 |
| | Current LW2 ($n_l = 1$) | 0.8302 |
| | Current LW2 ($n_l = 3$) | 0.8063 |
| | Current LW2 ($n_l = 6$) | 0.8062 |
| | Current LW3 ($n_l = 1$) | 0.8134 |
| | Current LW3 ($n_l = 3$) | 0.8061 |
| | Current LW3 ($n_l = 6$) | 0.8061 |
| | [a] Current LW3 ($n_l = 6$) | 0.8123 |
| | [b] Current LW3 ($n_l = 6$) | 0.8896 |
| | [a, b] Current LW3 ($n_l = 6$) | 0.8965 |
| | HSDT [80] | 0.81683 |
| | HRSDT [22] | 0.8833 |
| | TSDT [79] | 0.8832 |
| | FSDT [79] | 0.8868 |
| | EBT [79] | 1.0370 |
| 30 | Current LW1 ($n_l = 1$) | 0.7828 |
| | Current LW1 ($n_l = 3$) | 0.7622 |
| | Current LW1 ($n_l = 6$) | 0.7463 |
| | Current LW1 ($n_l = 9$) | 0.7432 |
| | Current LW2 ($n_l = 1$) | 0.7824 |
| | Current LW2 ($n_l = 3$) | 0.7411 |
| | Current LW2 ($n_l = 6$) | 0.7407 |
| | Current LW3 ($n_l = 1$) | 0.7454 |
| | Current LW3 ($n_l = 3$) | 0.7406 |
| | Current LW3 ($n_l = 6$) | 0.7406 |
| | [a] Current LW3 ($n_l = 6$) | 0.7445 |
| | [b] Current LW3 ($n_l = 6$) | 0.7680 |
| | [a, b] Current LW3 ($n_l = 6$) | 0.7720 |
| | HSDT [80] | 0.72608 |
| | HRSDT [22] | 0.7472 |
| | TSDT [79] | 0.7471 |
| | FSDT [79] | 0.7528 |
| | EBT [79] | 1.1329 |

[a] von Karman geometric nonlinear terms are considered. [b] The Poisson ratio is considered to be zero in $Q_{ij}$ ($i = 1$ and 3).

The results also show that the critical temperature change parameters obtained using the TI material properties always are greater than those obtained using the TD material properties. For a moderately thick beam ($L/h = 10$ and $\kappa_p = 1$), deviations between the critical temperature change parameters obtained using the TI and the TD material properties are as high as approximately 36.5%, such that the current analysis using TI material properties is an unsafe analysis and, thus, is not recommended. The effects of TD

material properties on the critical temperature change parameters are significant, and these must be considered for the current issue.

**Table 3.** Critical temperature parameter solutions for simply supported, FG beams under a uniform temperature change.

| L/h | $\kappa_p$ | Micromechanical Models | Theories | $\overline{\Delta T_{cr}}$(TI) | $\overline{\Delta T_{cr}}$(TD) |
|---|---|---|---|---|---|
| 10 | 1 | Mori–Tanaka | Current LW1 ($n_l = 2$) | 0.5151 | 0.3757 |
| | | | Current LW1 ($n_l = 4$) | 0.5043 | 0.3692 |
| | | | Current LW1 ($n_l = 8$) | 0.5015 | 0.3674 |
| | | | Current LW2 ($n_l = 2$) | 0.5007 | 0.3674 |
| | | | Current LW2 ($n_l = 4$) | 0.5006 | 0.3667 |
| | | | Current LW3 ($n_l = 2$) | 0.5006 | 0.3667 |
| | | | Current LW3 ($n_l = 4$) | 0.5006 | 0.3667 |
| 10 | 1 | Rule of mixtures | Current LW1 ($n_l = 2$) | 0.5122 | 0.3678 |
| | | | Current LW1 ($n_l = 4$) | 0.5014 | 0.3614 |
| | | | Current LW1 ($n_l = 8$) | 0.4987 | 0.3600 |
| | | | Current LW2 ($n_l = 2$) | 0.4978 | 0.3595 |
| | | | Current LW2 ($n_l = 4$) | 0.4978 | 0.3595 |
| | | | Current LW3 ($n_l = 2$) | 0.4977 | 0.3595 |
| | | | Current LW3 ($n_l = 4$) | 0.4977 | 0.3595 |
| 10 | 5 | Mori–Tanaka | Current LW1 ($n_l = 2$) | 0.5315 | 0.4525 |
| | | | Current LW1 ($n_l = 4$) | 0.5202 | 0.4436 |
| | | | Current LW1 ($n_l = 8$) | 0.5172 | 0.4412 |
| | | | Current LW2 ($n_l = 2$) | 0.5164 | 0.4405 |
| | | | Current LW2 ($n_l = 4$) | 0.5163 | 0.4405 |
| | | | Current LW3 ($n_l = 2$) | 0.5163 | 0.4405 |
| | | | Current LW3 ($n_l = 4$) | 0.5163 | 0.4405 |
| 10 | 5 | Rule of mixtures | Current LW1 ($n_l = 2$) | 0.5307 | 0.4467 |
| | | | Current LW1 ($n_l = 4$) | 0.5193 | 0.4381 |
| | | | Current LW1 ($n_l = 8$) | 0.5164 | 0.4358 |
| | | | Current LW2 ($n_l = 2$) | 0.5155 | 0.4350 |
| | | | Current LW2 ($n_l = 4$) | 0.5154 | 0.4350 |
| | | | Current LW3 ($n_l = 2$) | 0.5154 | 0.4350 |
| | | | Current LW3 ($n_l = 4$) | 0.5154 | 0.4350 |
| 20 | 5 | Mori–Tanaka | Current LW1 ($n_l = 2$) | 0.5432 | 0.5194 |
| | | | Current LW1 ($n_l = 4$) | 0.5322 | 0.5081 |
| | | | Current LW1 ($n_l = 8$) | 0.5295 | 0.5066 |
| | | | Current LW2 ($n_l = 2$) | 0.5286 | 0.5042 |
| | | | Current LW2 ($n_l = 4$) | 0.5285 | 0.5042 |
| | | | Current LW3 ($n_l = 2$) | 0.5285 | 0.5042 |
| | | | Current LW3 ($n_l = 4$) | 0.5285 | 0.5042 |
| 20 | 5 | Rule of mixtures | Current LW1 ($n_l = 2$) | 0.5423 | 0.5170 |
| | | | Current LW1 ($n_l = 4$) | 0.5314 | 0.5066 |
| | | | Current LW1 ($n_l = 8$) | 0.5286 | 0.5042 |
| | | | Current LW2 ($n_l = 2$) | 0.5277 | 0.5026 |
| | | | Current LW2 ($n_l = 4$) | 0.5277 | 0.5026 |
| | | | Current LW3 ($n_l = 2$) | 0.5277 | 0.5026 |
| | | | Current LW3 ($n_l = 4$) | 0.5277 | 0.5026 |

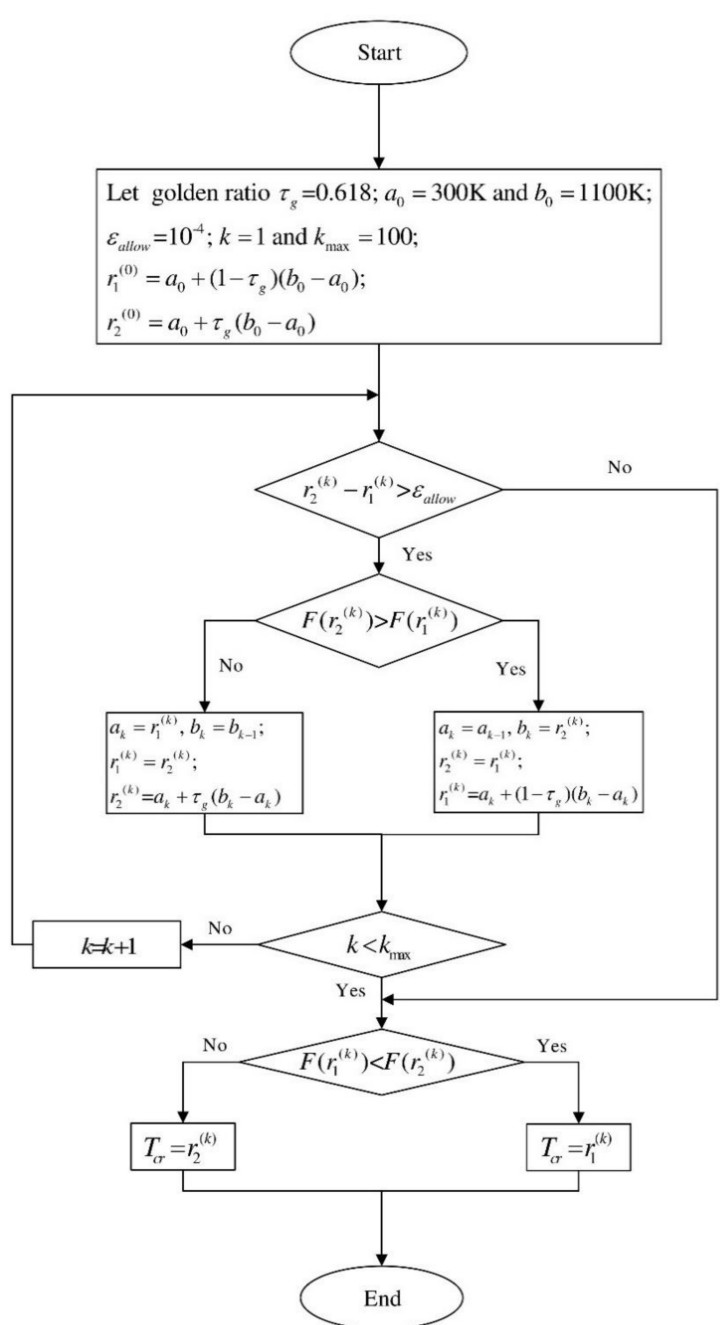

**Figure 5.** Flow chart of the golden section method used for the cases of temperature-dependent (TD) material properties.

### 5.2. Optimization of Material Composition of FG beams

In this section, a multiple-objective optimization for material composition of a simply-supported FG beam subjected to a uniform temperature change is considered in order to maximize its critical temperature change and minimize its self-weight. In the optimal design, the FG beam is considered to be a two-phase composite material, where one phase is the ceramic material ($ZrO_2$) and the other is the metal material (SUS304). The TD material properties of $ZrO_2$ and SUS304 are given in Table 4 [4]. Variations in the material properties of $ZrO_2$ and SUS304 with the temperature variable ranging from 300 K to 1100 K are shown in Figure 6. The material properties of the FG beam are assumed to obey a three-parameter power-law distribution of volume fractions of the constituents along the thickness of the FG beam, and the effective material properties are estimated using the rule of mixtures. The non-dominated sorting-based GA mentioned in Section 4.2 is used to determine some

sets of Pareto-optimal solutions of the undetermined coefficients $\kappa_p$, $\kappa_b$, and $\kappa_c$, in which 200 initial populations are randomly generated, in which the ranges of $\kappa_p$, $\kappa_b$, and $\kappa_c$ are taken as $0 \leq \kappa_p \leq 50$, $0 \leq \kappa_b \leq 1$, and $1 \leq \kappa_c \leq 3$, respectively.

**Table 4.** Temperature dependent material properties of the metal material (SUS304) and the ceramic material (ZrO$_2$) [4].

| Materials | $P_0$ | $P_{-1}$ | $P_1$ | $P_2$ | $P_3$ | $P$ at 300K |
|---|---|---|---|---|---|---|
| **ZrO$_2$** | | | | | | |
| $E$ | $244.27 \times 10^9$ | 0 | $-1.371 \times 10^{-3}$ | $1.214 \times 10^{-6}$ | $-3.681 \times 10^{-10}$ | $168.06 \times 10^9$ |
| $\alpha$ | $12.766 \times 10^{-6}$ | 0 | $-1.491 \times 10^{-3}$ | $1.006 \times 10^{-5}$ | $-6.778 \times 10^{-11}$ | $18.591 \times 10^{-6}$ |
| $\upsilon$ | 0.2882 | 0 | $1.133 \times 10^{-4}$ | 0 | 0 | 0.298 |
| $\rho$ | 3657 | 0 | 0 | 0 | 0 | 3657 |
| **SUS304** | | | | | | |
| $E$ | $201.04 \times 10^9$ | 0 | $3.079 \times 10^{-4}$ | $-6.534 \times 10^{-7}$ | 0 | $207.79 \times 10^9$ |
| $\alpha$ | $12.330 \times 10^{-6}$ | 0 | $8.086 \times 10^{-4}$ | 0 | 0 | $15.321 \times 10^{-6}$ |
| $\upsilon$ | 0.3262 | 0 | $-2.002 \times 10^{-4}$ | $3.797 \times 10^{-7}$ | 0 | 0.318 |
| $\rho$ | 8166 | 0 | 0 | 0 | 0 | 8166 |

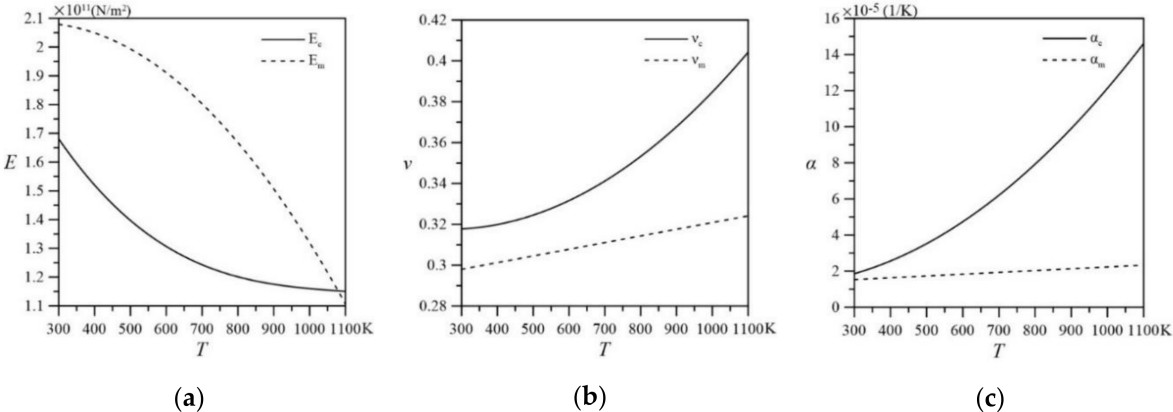

**Figure 6.** Variations of material properties of the ceramic and the metal materials with the temperature: (**a**) Young's modulus; (**b**) Poisson's ratio; (**c**) Thermal expansion coefficient.

Figures 7 and 8 show the populations at the initial, first, second, fifth, tenth, and 20th generations, where the TI and TD material properties are considered, respectively. It can be seen in Figures 7 and 8 that the non-dominated, sorting-based GA converges rapidly and that the Poreto-optimal solutions can be yielded after 20 generations. Curve fitting for these Preto-optimal solutions is performed for the cases of TI and TD material properties, which is shown in Figure 9 with the solid and dashed lines, respectively, in which

$$F_2 = -1.3303(F_1)^4 + 1.1843(F_1)^3 + 0.4537(F_1)^2 - 1.3(F_1) + 1.0005 \text{ for the TI material property cases,} \quad (68)$$

$$F_2 = -1.8376(F_1)^4 + 2.4524(F_1)^3 - 1.2222(F_1)^2 - 0.3734(F_1) + 0.9936 \text{for the TD material property cases.} \quad (69)$$

Equations (68) and (69) can then be used to calculate the associated weight numbers (i.e., $w_1$ and $w_2$) for the objective functions $F_1$ and $F_2$, in which if we let $dF_2/dF_1 = \lambda$, then $w_1 = -\lambda/(1-\lambda)$ and $w_2 = 1/(1-\lambda)$.

Tables 5 and 6 show every ten Pareto-optimal solutions sorted by the $F_2$ function values from the largest to the smallest for the TI and TD material property cases, respectively. These Pareto-optimal solutions may provide design engineers with valuable information regarding what set of the material-property gradient indices ($\kappa_p$, $\kappa_b$, and $\kappa_c$) they need according to the weight number ratio of ($w_2/w_1$). In addition, the results also show that the critical temperature changes of the FG beam for the TD material property cases are much less than those for the TI material property cases, such that the TD material property effects must be considered in TD problems.

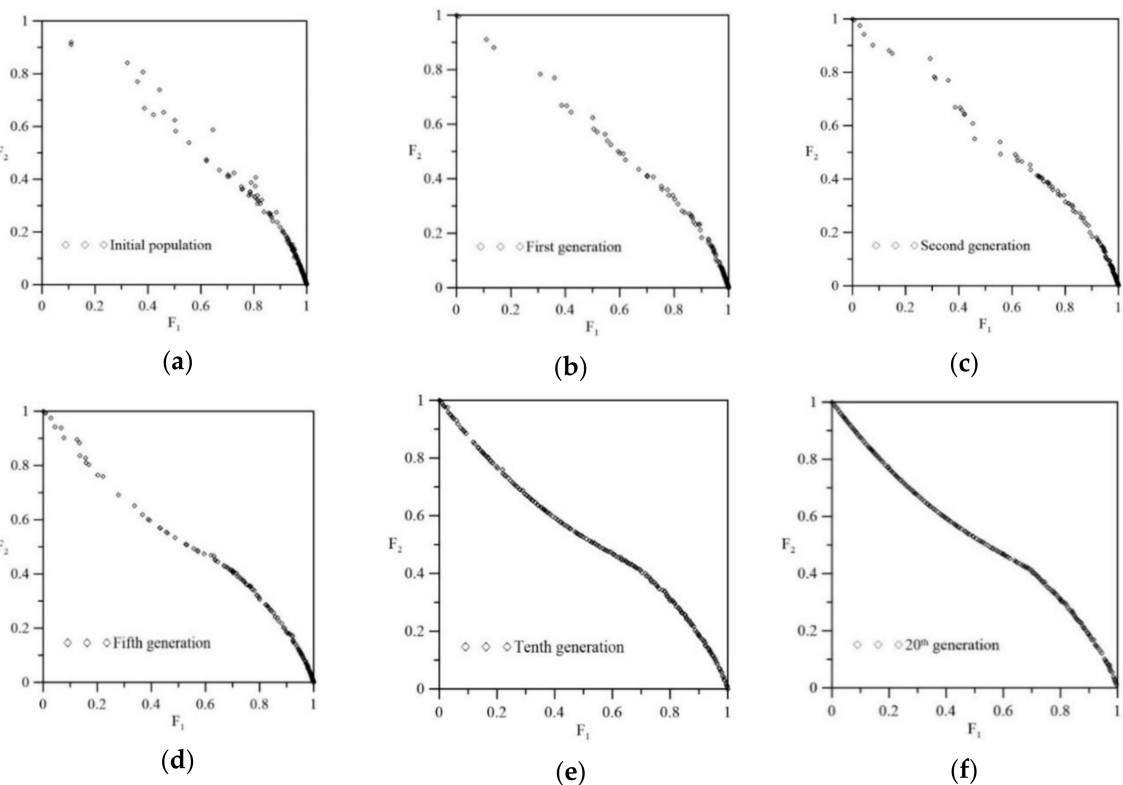

**Figure 7.** Two hundred populations for different generations shown in the objective spaces, in which the temperature-independent (TI) material properties are considered; (**a**) initial generation; (**b**) 1st generation; (**c**) 2nd generation; (**d**) 5th generation; (**e**) 10th generation; (**f**) 20th generation.

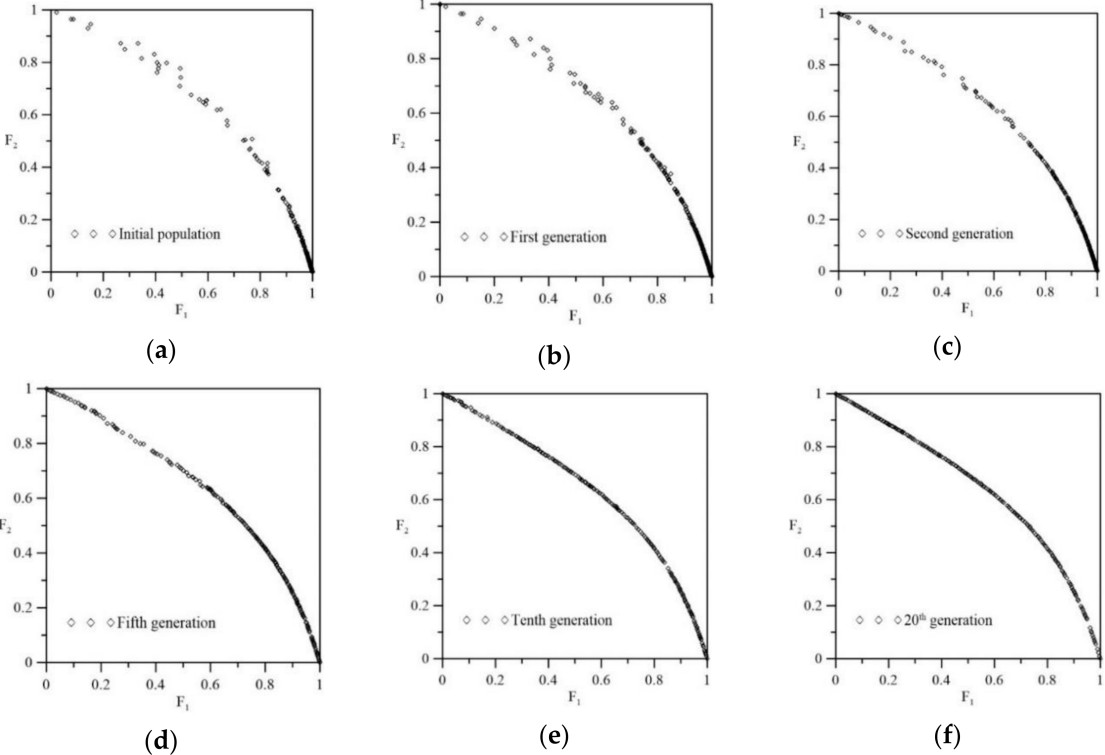

**Figure 8.** Two hundred populations for different generations shown in the objective spaces, in which the TD material properties are considered; (**a**) initial generation; (**b**) 1st generation; (**c**) 2nd generation; (**d**) 5th generation; (**e**) 10th generation; (**f**) 20th generation.

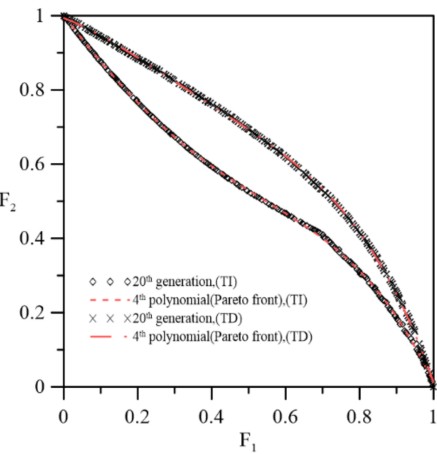

**Figure 9.** Curve fitting for the Pareto-optimal fronts for the cases of TI and TD material properties.

**Table 5.** Pareto-optimal solutions for simply supported FG beams under a uniform temperature change sorted according to the $F_2$ function values from the largest to the smallest, in which temperature-independent material properties are considered.

| Order Number of Pareto-Optimal Solutions | $(F_1, F_2)$ | $\hat{\rho}$ | $\Delta T_{cr}$ | $(\kappa_b, \kappa_c, \kappa_p)$ | $(w_1, w_2)$ |
|---|---|---|---|---|---|
| 1 | (0.000, 1.000) | 3658.1 | 300.7326 | (0.358, 1.829, 0.000) | (0.565, 0.435) |
| 10 | (0.037, 0.953) | 3820.7 | 303.1011 | (0.000, 1.000, 0.037) | (0.558, 0.442) |
| 20 | (0.072, 0.910) | 3975.6 | 305.2840 | (0.000, 2.392, 0.076) | (0.549, 0.451) |
| 30 | (0.113, 0.861) | 4157.9 | 307.7473 | (0.000, 3.000, 0.125) | (0.537, 0.463) |
| 40 | (0.154, 0.815) | 4337.3 | 310.0472 | (0.000, 2.831, 0.178) | (0.523, 0.477) |
| 50 | (0.206, 0.760) | 4565.6 | 312.7782 | (0.000, 2.759, 0.253) | (0.502, 0.498) |
| 60 | (0.251, 0.716) | 4768.5 | 315.0162 | (0.000, 2.714, 0.328) | (0.482, 0.518) |
| 70 | (0.309, 0.665) | 5023.2 | 317.5743 | (0.000, 3.000, 0.435) | (0.456, 0.544) |
| 80 | (0.380, 0.610) | 5335.0 | 320.3524 | (0.000, 2.967, 0.591) | (0.424, 0.576) |
| 90 | (0.433, 0.572) | 5570.0 | 322.2288 | (0.002, 2.958, 0.736) | (0.403, 0.597) |
| 100 | (0.496, 0.531) | 5851.2 | 324.2938 | (0.001, 2.958, 0.945) | (0.385, 0.615) |
| 110 | (0.576, 0.482) | 6204.6 | 326.7625 | (0.004, 2.820, 1.300) | (0.381, 0.619) |
| 120 | (0.644, 0.439) | 6502.7 | 328.9122 | (0.000, 3.000, 1.723) | (0.399, 0.601) |
| 130 | (0.695, 0.403) | 6731.2 | 330.7254 | (0.000, 2.745, 2.190) | (0.428, 0.572) |
| 140 | (0.741, 0.367) | 6934.5 | 332.5478 | (0.653, 1.122, 7.805) | (0.459, 0.541) |
| 150 | (0.801, 0.311) | 7199.1 | 335.3457 | (0.864, 1.065, 22.95) | (0.506, 0.494) |
| 160 | (0.844, 0.264) | 7387.2 | 337.7239 | (0.864, 1.082, 25.69) | (0.545, 0.455) |
| 170 | (0.883, 0.213) | 7559.9 | 340.2622 | (0.881, 1.179, 24.36) | (0.580, 0.420) |
| 180 | (0.926, 0.148) | 7751.7 | 343.5581 | (0.954, 1.146, 50.00) | (0.621, 0.379) |
| 190 | (0.967, 0.076) | 7930.0 | 347.1474 | (0.537, 1.000, 45.04) | (0.661, 0.339) |
| 200 | (1.000, 0.008) | 8077.5 | 350.5544 | (0.000, 2.005, 50.00) | (0.684, 0.316) |

**Table 6.** Pareto-optimal solutions of simply supported FG beams under a uniform temperature change sorted according to the $F_2$ function values from the largest to the smallest, in which temperature-dependent material properties are considered.

| Order Number of Pareto-Optimal Solutions | $(F_1, F_2)$ | $\hat{\rho}$ | $\Delta T_{cr}$ | $(\kappa_b, \kappa_c, \kappa_p)$ | $(w_1, w_2)$ |
|---|---|---|---|---|---|
| 1 | (0.000, 0.994) | 3657.0 | 172.5 | (0.465, 1.620, 0.000) | (0.272, 0.728) |
| 10 | (0.040, 0.977) | 3833.8 | 174.4 | (0.000, 3.000, 0.041) | (0.315, 0.685) |
| 20 | (0.098, 0.947) | 4090.2 | 177.7 | (0.000, 3.000, 0.105) | (0.354, 0.646) |
| 30 | (0.151, 0.917) | 4326.6 | 181.1 | (0.000, 1.799, 0.173) | (0.375, 0.625) |
| 40 | (0.199, 0.888) | 4535.2 | 184.4 | (0.000, 1.787, 0.241) | (0.385, 0.615) |
| 50 | (0.246, 0.858) | 4743.0 | 187.7 | (0.000, 3.000, 0.318) | (0.390, 0.610) |
| 60 | (0.311, 0.816) | 5031.9 | 192.4 | (0.000, 1.000, 0.440) | (0.391, 0.609) |
| 70 | (0.364, 0.781) | 5268.3 | 196.2 | (0.000, 2.613, 0.560) | (0.391, 0.609) |

**Table 6.** *Cont.*

| Order Number of Pareto-Optimal Solutions | $(F_1, F_2)$ | $\hat{\rho}$ | $\Delta T_{cr}$ | $(\kappa_b, \kappa_c, \kappa_p)$ | $(w_1, w_2)$ |
|---|---|---|---|---|---|
| 80 | (0.424, 0.743) | 5532.8 | 200.6 | (0.000, 1.561, 0.716) | (0.393, 0.607) |
| 90 | (0.483, 0.705) | 5791.1 | 204.8 | (0.000, 1.474, 0.901) | (0.400, 0.600) |
| 100 | (0.531, 0.672) | 6002.2 | 208.5 | (0.000, 2.792, 1.086) | (0.411, 0.589) |
| 110 | (0.585, 0.633) | 6242.9 | 212.9 | (0.013, 1.444, 1.364) | (0.431, 0.569) |
| 120 | (0.651, 0.579) | 6533.8 | 218.8 | (0.000, 2.239, 1.746) | (0.465, 0.535) |
| 130 | (0.697, 0.537) | 6736.6 | 223.6 | (0.000, 1.443, 2.132) | (0.496, 0.504) |
| 140 | (0.746, 0.484) | 6956.2 | 229.5 | (0.046, 2.894, 2.754) | (0.535, 0.465) |
| 150 | (0.793, 0.425) | 7164.5 | 236.1 | (0.343, 1.047, 5.655) | (0.575, 0.425) |
| 160 | (0.836, 0.362) | 7353.8 | 243.1 | (0.373, 1.592, 5.894) | (0.612, 0.388) |
| 170 | (0.884, 0.281) | 7562.9 | 252.1 | (0.428, 1.210, 10.03) | (0.652, 0.348) |
| 180 | (0.915, 0.219) | 7703.5 | 259.1 | (0.798, 1.231, 21.23) | (0.677, 0.323) |
| 190 | (0.965, 0.105) | 7924.0 | 271.9 | (0.879, 1.297, 34.51) | (0.713, 0.287) |
| 200 | (1.000, 0.013) | 8077.6 | 282.1 | (0.117, 1.000, 49.74) | (0.736, 0.264) |

## 6. Conclusions

In this work, the authors developed a mixed LW HSDT for the thermal buckling analysis of FG beams subjected to a uniform temperature change, and then they further developed a non-dominated sorting-based GA for multi-objectives optimization of the material composition of a three-parameter FG beam, in which the TI and TD material properties are considered. All of the results in the tables and figures are obtained using the self-developed MATLAB programs without any commercial toolbox.

In the thermal buckling analysis, the results show that the TD material properties must be considered in TD physical problems, in which for a moderately thick FG beam, the deviations in the critical temperature change parameters obtained using the TD material properties and the TI material properties are as high as approximately 36.5%, and the analysis based on the TI material materials is an unsafe analysis, due to the fact that it always overestimates the actual critical temperature change parameters.

In the optimal design cases, the results show the self-developed non-dominated sorting-based GA converges rapidly, where the Poreto-optimal solutions can be yielded after the 20th generation. The non-dominated sorting-based GA can also be extended to other optimal designs of FG beams with multiple objective functions.

**Author Contributions:** Conceptualization, C.-P.W.; methodology, C.-P.W.; software, K.-W.L.; validation, C.-P.W. and K.-W.L.; investigation: C.-P.W. and K.-W.L.; resources, C.-P.W.; data curation, K.-W.L.; writing—original draft preparation, C.-P.W.; writing—review and editing, C.-P.W.; visualization, C.-P.W.; supervision, C.-P.W.; project administration, C.-P.W. funding acquisition, C.-P.W. All authors have read and agreed to the published version of the manuscript.

**Funding:** This research was funded by the Ministry of Science and Technology, Taiwan, grant number MOST 109-2221-E-006-015-MY3.

**Institutional Review Board Statement:** This study did not involve humans and animals.

**Informed Consent Statement:** This study did not involve humans and animals.

**Data Availability Statement:** All of the data for Figures 1–9 can be obtained at https://drive.google.com/drive/folders/1J2KY29JdaCAhlpuGJdLSz0fb3kU_Z-HP?usp=sharing (accessed on 30 March 2021).

**Conflicts of Interest:** The authors declare no conflict of interest.

## Appendix A. Relations between the Generalized Force/Moment Resultants and the Generalized Displacements

The definition of generalized force and moment resultants and their relations with the displacement components are given as follows:

$$
\begin{aligned}
N_x^{(m)} &= \int_{-h_m/2}^{h_m/2} \sigma_x^{(m)} dz_m \\
&= A_{11}^{(m)} u_0^{(m)},_x + B_{11}^{(m)} u_1^{(m)},_x + D_{11}^{(m)} u_2^{(m)},_x + F_{11}^{(m)} u_3^{(m)},_x + A_{13}^{(m)} w_1^{(m)} + 2B_{13}^{(m)} w_2^{(m)} + 3D_{13}^{(m)} w_3^{(m)},
\end{aligned}
\tag{A1}
$$

$$
\begin{aligned}
M_x^{(m)} &= \int_{-h_m/2}^{h_m/2} \sigma_x^{(m)} z_m \, dz_m \\
&= B_{11}^{(m)} u_0^{(m)},_x + D_{11}^{(m)} u_1^{(m)},_x + F_{11}^{(m)} u_2^{(m)},_x + H_{11}^{(m)} u_3^{(m)},_x + B_{13}^{(m)} w_1^{(m)} + 2D_{13}^{(m)} w_2^{(m)} + 3F_{13}^{(m)} w_3^{(m)},
\end{aligned}
\tag{A2}
$$

$$
\begin{aligned}
P_x^{(m)} &= \int_{-h_m/2}^{h_m/2} \sigma_x^{(m)} z_m^2 \, dz_m \\
&= D_{11}^{(m)} u_0^{(m)},_x + F_{11}^{(m)} u_1^{(m)},_x + H_{11}^{(m)} u_2^{(m)},_x + J_{11}^{(m)} u_3^{(m)},_x + D_{13}^{(m)} w_1^{(m)} + 2F_{13}^{(m)} w_2^{(m)} + 3H_{13}^{(m)} w_3^{(m)},
\end{aligned}
\tag{A3}
$$

$$
\begin{aligned}
R_x^{(m)} &= \int_{-h_m/2}^{h_m/2} \sigma_x^{(m)} z_m^3 \, dz_m \\
&= F_{11}^{(m)} u_0^{(m)},_x + H_{11}^{(m)} u_1^{(m)},_x + J_{11}^{(m)} u_2^{(m)},_x + L_{11}^{(m)} u_3^{(m)},_x + F_{13}^{(m)} w_1^{(m)} + 2H_{13}^{(m)} w_2^{(m)} + 3J_{13}^{(m)} w_3^{(m)},
\end{aligned}
\tag{A4}
$$

$$
\begin{aligned}
N_z^{(m)} &= \int_{-h_m/2}^{h_m/2} \sigma_z^{(m)} dz_m \\
&= A_{13}^{(m)} u_0^{(m)},_x + B_{13}^{(m)} u_1^{(m)},_x + D_{13}^{(m)} u_2^{(m)},_x + F_{13}^{(m)} u_3^{(m)},_x + A_{33}^{(m)} w_1^{(m)} + 2B_{33}^{(m)} w_2^{(m)} + 3D_{33}^{(m)} w_3^{(m)},
\end{aligned}
\tag{A5}
$$

$$
\begin{aligned}
M_z^{(m)} &= \int_{-h_m/2}^{h_m/2} \sigma_z^{(m)} z_m \, dz_m \\
&= B_{13}^{(m)} u_0^{(m)},_x + D_{13}^{(m)} u_1^{(m)},_x + F_{13}^{(m)} u_2^{(m)},_x + H_{13}^{(m)} u_3^{(m)},_x + B_{33}^{(m)} w_1^{(m)} + 2D_{33}^{(m)} w_2^{(m)} + 3F_{33}^{(m)} w_3^{(m)},
\end{aligned}
\tag{A6}
$$

$$
\begin{aligned}
P_z^{(m)} &= \int_{-h_m/2}^{h_m/2} \sigma_z^{(m)} z_m^2 \, dz_m \\
&= D_{13}^{(m)} u_0^{(m)},_x + F_{13}^{(m)} u_1^{(m)},_x + H_{13}^{(m)} u_2^{(m)},_x + J_{13}^{(m)} u_3^{(m)},_x + D_{33}^{(m)} w_1^{(m)} + 2F_{33}^{(m)} w_2^{(m)} + 3H_{33}^{(m)} w_3^{(m)},
\end{aligned}
\tag{A7}
$$

$$
\begin{aligned}
N_{xz}^{(m)} &= \int_{-h_m/2}^{h_m/2} \tau_{xz}^{(m)} dz_m \\
&= A_{55}^{(m)} \left( w_0^{(m)},_x + u_1^{(m)} \right) + B_{55}^{(m)} \left( w_1^{(m)},_x + 2u_2^{(m)} \right) + D_{55}^{(m)} \left( w_2^{(m)},_x + 3u_3^{(m)} \right) + F_{55}^{(m)} w_3^{(m)},_x,
\end{aligned}
\tag{A8}
$$

$$
\begin{aligned}
M_{xz}^{(m)} &= \int_{-h_m/2}^{h_m/2} \tau_{xz}^{(m)} z_m \, dz_m \\
&= B_{55}^{(m)} \left( w_0^{(m)},_x + u_1^{(m)} \right) + D_{55}^{(m)} \left( w_1^{(m)},_x + 2u_2^{(m)} \right) + F_{55}^{(m)} \left( w_2^{(m)},_x + 3u_3^{(m)} \right) + H_{55}^{(m)} w_3^{(m)},_x,
\end{aligned}
\tag{A9}
$$

$$
\begin{aligned}
P_{xz}^{(m)} &= \int_{-h_m/2}^{h_m/2} \tau_{xz}^{(m)} z_m^2 \, dz_m \\
&= D_{55}^{(m)} \left( w_0^{(m)},_x + u_1^{(m)} \right) + F_{55}^{(m)} \left( w_1^{(m)},_x + 2u_2^{(m)} \right) + H_{55}^{(m)} \left( w_2^{(m)},_x + 3u_3^{(m)} \right) + J_{55}^{(m)} w_3^{(m)},_x,
\end{aligned}
\tag{A10}
$$

$$
\begin{aligned}
R_{xz}^{(m)} &= \int_{-h_m/2}^{h_m/2} \tau_{xz}^{(m)} z_m^3 dz_m \\
&= F_{55}^{(m)} \left( w_0^{(m)},_x + u_1^{(m)} \right) + H_{55}^{(m)} \left( w_1^{(m)},_x + 2u_2^{(m)} \right) + J_{55}^{(m)} \left( w_2^{(m)},_x + 3u_3^{(m)} \right) + L_{55}^{(m)} w_3^{(m)},_x,
\end{aligned}
\tag{A11}
$$

where

$$
\left\{ A_{ij}^{(m)} \quad B_{ij}^{(m)} \quad D_{ij}^{(m)} \quad F_{ij}^{(m)} \quad H_{ij}^{(m)} \quad J_{ij}^{(m)} \quad L_{ij}^{(m)} \right\}^T = \int_{-h_m/2}^{h_m/2} Q_{ij}^{(m)} \left\{ 1 \quad z_m \quad z_m^2 \quad z_m^3 \quad z_m^4 \quad z_m^5 \quad z_m^6 \right\}^T dz_m,
$$

in which the superscript *T* refers to the transport of the vector.

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
