# Peer review of "Multi-Objective Optimization of Functionally Graded Beams Using a Genetic Algorithm with Non-Dominated Sorting"

_jcs, doi:10.3390/jcs5040092_

Round 1
Reviewer 1 Report
The authors focus on investigating the material composition optimization of an FG beam under a uniform temperature change in order to maximize the critical temperature change parameter and minimize the total mass of the FG beam. This manuscript is well written. However, the authors need to improve this manuscript in many areas. The following are the points that need to be addressed.
Recommendation: Major revision
1. One would easily be lost in the long equations. The long equations can be shorted using the assigned variables, especially equations 14 and 15.
2. line 152: the authors used "Fig." instead of "Figure". Change this throughout the manuscript.
3. Line 323: Why subtopic (b) appeared there? Are you missing (a)?
4. In table 4, the material properties are obtained from Ref. 4. Put this reference in the table caption.
5. In this work, the authors mainly focused on the observations. However, it would help the reader to report the critical assessment of obtained results.
6. The conclusion of the manuscript can be shortened.
Reviewer 2 Report
The manuscript presented a mixed layerwise and higher-order shear deformation model for thermal instability analysis of functionally graded beams. Based on the developed model, a multi-objective optimization analysis is also carried out to maximize the critical temperature change parameters and minimize the total mass according to a non-dominated sorting based genetic algorithm. The manuscript is well written, and the topic falls within the scope of the journal. Before recommending publication, I would like to ask the authors to address the remarks listed below.
(1) Do eqs. (7) and (8) mean that the displacement is C0 continuous at the ply interface? In other words, the transverse strain components will be discontinuous at the interface. If so, it needs to be clearly explained.
(2) Is the proposed FG beam model a 2D model or a 3D model? I don’t see the y components of the stresses and strains (i.e., xy, yy, yz) in eq. (12).
(3) Since the manuscript is concerned with the thermal buckling analysis of FG beams, how did the authors capture the loading shedding instability behavior? I can’t find the corresponding discussion in the manuscript. Is the post-buckling response (where the stiffness becomes negative) considered in this work?
(4) The authors are suggested to briefly mention a few recent publications on the modeling of functionally graded materials and thermal buckling analysis: (a) doi.org/10.1016/j.compstruct.2017.05.037, (b) doi.org/10.1016/j.compstruct.2020.111893, (c) doi.org/10.1061/(ASCE)EM.1943-7889.0001263, (d) doi.org/10.1080/10407790.2019.1627801
Round 2
Reviewer 1 Report
Accept in the present form.
Reviewer 2 Report
A relevant and important paper on the modeling of FG structures is left out and should be included: doi.org/10.1016/j.compstruct.2017.05.037